# Meta-Learning Adaptive Loss Functions

**Christian Raymond**                                       *christian.raymond@ecs.vuw.ac.nz*
*Victoria University of Wellington*

**Qi Chen**                                                       *qi.chen@ecs.vuw.ac.nz*
*Victoria University of Wellington*

**Bing Xue**                                                     *bing.xue@ecs.vuw.ac.nz*
*Victoria University of Wellington*

**Mengjie Zhang**                                          *mengjie.zhang@ecs.vuw.ac.nz*
*Victoria University of Wellington*

**Reviewed on OpenReview:** *https://openreview.net/forum?id=oOODnNOxz8*

## Abstract

Loss function learning is a new meta-learning paradigm that aims to automate the essential task of designing a loss function for a machine learning model. Existing techniques for loss function learning have shown promising results, often improving a model's training dynamics and final inference performance. However, a significant limitation of these techniques is that the loss functions are meta-learned in an offline fashion, where the meta-objective only considers the very first few steps of training, which is a significantly shorter time horizon than the one typically used for training deep neural networks. This causes significant bias towards loss functions that perform well at the very start of training but perform poorly at the end of training. To address this issue we propose a new loss function learning technique for adaptively updating the loss function online after each update to the base model parameters. The experimental results show that our proposed method consistently outperforms the cross-entropy loss and offline loss function learning techniques on a diverse range of neural network architectures and datasets.

## 1 Introduction

When applying deep neural networks to a given learning task, a significant amount of time is typically allocated towards performing manual tuning of the hyper-parameters to achieve competitive learning performances (Bengio, 2012). Selection of the appropriate hyper-parameters is critical for embedding the relevant inductive biases into the learning algorithm (Gordon & Desjardins, 1995). The inductive biases control both the set of searchable models and the learning rules used to find the final model parameters. Therefore, the field of meta-learning (Schmidhuber, 1987; Vanschoren, 2018; Peng, 2020; Hospedales et al., 2022), as well as the closely related field of hyper-parameter optimization (Bergstra et al., 2011; Feurer & Hutter, 2019), aim to automate the design and selection of a suitable set of inductive biases (or a subset of them) and have been long-standing areas of interest to the machine learning community.

One component that has only very recently been receiving attention in the meta-learning context is the loss function. The loss function (Wang et al., 2022) is one of the most central components of any gradient-based supervised learning system, as it determines the base learning algorithm's learning path and the selection of the final model (Reed & MarksII, 1999). Furthermore, in deep learning, neural networks are typically trained through the backpropagation of gradients that originate from the loss function (Rumelhart et al., 1986; Goodfellow et al., 2016). Given this importance, a new and emerging subfield of meta-learning referred to as *Loss Function Learning* (Gonzalez & Miikkulainen, 2020b; Bechtle et al., 2021; Raymond et al., 2023; Collet et al., 2022) aims to attempt the difficult task of inferring a highly performant loss function directly from the given data.

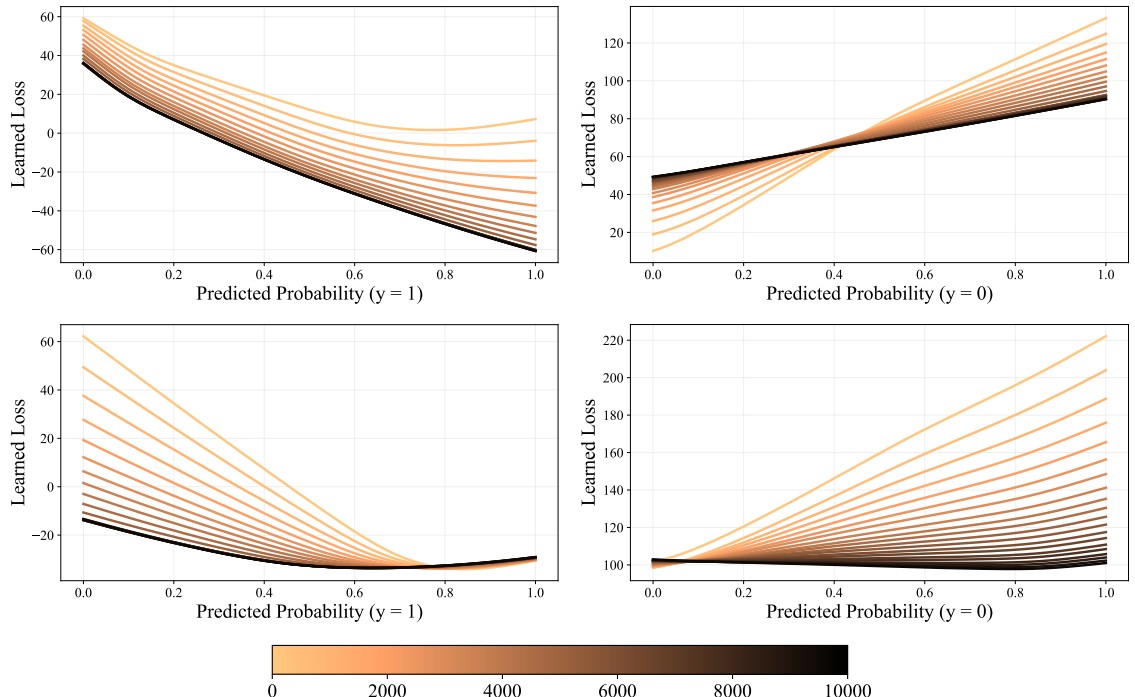

Figure 1: Example adaptive meta-learned loss functions generated by AdaLFL on the CIFAR-10 dataset, where each row represents a classification loss function, and the color represents the current gradient step.

Loss function learning aims to meta-learn a task-specific loss function, which yields improved performance capabilities when utilized in training compared to handcrafted loss functions. Initial approaches to loss function learning have shown promise at enhancing various aspects of deep neural network training, such as improving the convergence and sample efficiency (Gonzalez & Miikkulainen, 2020b; Bechtle et al., 2021), as well as the generalization (Gonzalez & Miikkulainen, 2021; Liu et al., 2020; Li et al., 2022; Leng et al., 2022), and model robustness (Gao et al., 2021; 2022). However, *one prevailing limitation* of the existing approaches to loss function learning is that they have thus far exclusively focused on learning a loss function in the offline meta-learning settings.

In offline loss function learning, training is prototypically partitioned into two phases. In the first phase, the base loss function is meta-learned via iteratively updating the loss function by performing one or a few base training steps to approximate the performance. Second, the base model is trained using the learned loss function, which is now *fixed* and is used in place of the conventional handcrafted loss function. Unfortunately, this methodology is prone to a severe short-horizon bias (Wu et al., 2018) towards loss functions which are performant in the early stages of training but often have poor performance in the later stages.

To address the limitation of offline loss function learning, we propose a new technique for *online* loss function learning called *Adaptive Loss Function Learning* (AdaLFL). In the proposed technique, the learned loss function is represented as a small feed-forward neural network trained simultaneously with the base learning model. Unlike prior methods, AdaLFL can adaptively transform both the shape and scale of the loss function throughout the learning process to adapt to what is required at each stage of the learning process, as shown in Figure 1. In offline loss function learning, the central goal is to improve the performance of a model by specializing the loss function to a small set of related tasks. Online loss function learning naturally extends this general philosophy, specializing the loss function to each individual gradient step on a single task.

## 1.1 Contributions

- We introduce a method for efficiently learning general-purpose adaptive loss functions using online meta-learning, where the loss function is updated after each base model update via a one-step unrolled differentiation algorithm.

- We identify and address shortcomings in the design of neural network-based loss function parameterizations, which previously caused learned loss functions to be biased toward overly flat shapes resulting in poor training dynamics.

- Empirically we show that models trained with our method exhibit faster convergence and improved inference performance compared to those trained with handcrafted or offline-learned loss functions.

- Finally, we analyze the learned loss functions and uncover key insights, such as implicit meta-learning, which reveals why meta-learned loss functions consistently outperform traditional handcrafted losses.

# 2 Online Loss Function Learning

In this work, we aim to automate the design and selection of the loss function and improve upon the performance of supervised machine learning systems. This is achieved via meta-learning an adaptive loss function that transforms both its shape and scale throughout the learning process. To achieve this, we propose *Adaptive Loss Function Learning* (AdaLFL), an efficient task and model-agnostic approach for online adaptation of the base loss function.

## 2.1 Problem Setup

In a prototypical supervised learning setup, we are given a set of $N$ independently and identically distributed (i.i.d.) examples of form $\mathcal{D} = \{(x_1, y_1), \ldots, (x_N, y_N)\}$, where $x_i \in X$ is the $i$th instance's feature vector and $y_i \in Y$ is its corresponding class label. We want to learn a mapping between $X$ and $Y$ using some base learning model, *e.g.*, a classifier or regressor, $f_\theta : \mathcal{X} \to \mathcal{Y}$, where $\theta$ is the base model parameters. In this paper, similar to others (Finn et al., 2017; Bechtle et al., 2021), we constrain the selection of the base models to those amenable to a stochastic gradient descent (SGD) style training procedures such that optimization of the model parameters $\theta$ occurs via optimizing some task-specific loss function $\mathcal{L}_\mathcal{T}$ with learning rule

$$\theta_{t+1} = \theta_t - \alpha \nabla_{\theta_t} \mathcal{L}_\mathcal{T}(y, f_{\theta_t}(x)), \tag{1}$$

where $\mathcal{L}_\mathcal{T}$ is a handcrafted loss function, typically the cross-entropy between the predicted label and the ground truth label in classification or the squared error in regression. The principal goal of AdaLFL is to reduce the reliance of models on manually designed handcrafted loss functions such as $\mathcal{L}_\mathcal{T}$, by instead using a meta-learned adaptive loss function $\mathcal{M}_\phi$, where the meta-parameters $\phi$ are learned simultaneously with the base parameters $\theta$, allowing for online adaptation of the loss function. We formulate the task of learning $\phi$ and $\theta$ as a non-stationary bi-level optimization problem, where $t$ is the current time step

$$\phi_{t+1} = \arg\min_\phi \mathcal{L}_\mathcal{T}(y, f_{\theta_{t+1}}(x))$$
$$s.t. \quad \theta_{t+1}(\phi_t) = \arg\min_\theta \mathcal{M}_{\phi_t}(y, f_{\theta_t}(x)). \tag{2}$$

The outer optimization aims to meta-learn a performant loss function $\mathcal{M}_\phi$ that minimizes the error on the given task. The inner optimization directly minimizes the learned loss value produced by $\mathcal{M}_\phi$ to learn the base model parameters $\theta$.

## 2.2 Loss Function Representation

In AdaLFL, the choice of loss function parameterization is a small feedforward neural network, which is chosen due to its high expressiveness and design flexibility. Our meta-learned loss function parameterization inspired by (Bechtle et al., 2021; Psaros et al., 2022) is a small feedforward neural network denoted by $\ell_\phi$

---

**Algorithm 1** Loss Function Initialization (Offline)

---

**Input:** $\mathcal{L}_{\mathcal{T}} \leftarrow$ Task loss function (meta-objective)

---

1: $\mathcal{M}_{\phi_0} \leftarrow$ Initialize parameters of meta learner
2: **for** $t \in \{0, ..., \mathcal{S}_{init} - 1\}$ **do**
3:     $\theta_0 \leftarrow$ Reset parameters of base learner
4:     **for** $i \in \{0, ..., \mathcal{S}_{inner} - 1\}$ **do**
5:       $X, y \leftarrow$ Sample from $\mathcal{D}_{train}$
6:       $\mathcal{M}_{learned} \leftarrow \mathcal{M}_{\phi_t}(y, f_{\theta_i}(X))$
7:       $\theta_{i+1} \leftarrow \theta_i - \alpha \nabla_{\theta_i} \mathcal{M}_{learned}$
8:     **end for**
9:     $X, y \leftarrow$ Sample from $\mathcal{D}_{valid}$
10:    $\mathcal{L}_{task} \leftarrow \mathcal{L}_{\mathcal{T}}(y, f_{\theta_{i+1}}(X))$
11:    $\phi_{t+1} \leftarrow \phi_t - \eta \nabla_{\phi_t} \mathcal{L}_{task}$
12: **end for**

---

with two hidden layers and 40 hidden units each, which is applied output-wise across $\mathcal{C}$ outputs (making it invariant to the number of outputs).

$$\mathcal{M}_{\phi}(y, f_{\theta}(x)) = \frac{1}{\mathcal{C}} \sum_{i=0}^{\mathcal{C}} \ell_{\phi}(y_i, f_{\theta}(x)_i) \tag{3}$$

Crucially, key design decisions are made regarding the activation functions used in $\ell_{\phi}$ to enforce desirable behavior. In Bechtle et al. (2021), ReLU activations are used in the hidden layers, and the smooth Softplus activation is used in the output layer to constrain the loss to be non-negative, *i.e.*, $\ell_{\phi} : \mathbb{R}^2 \to \mathbb{R}_0^+$. Unfortunately, this network architecture is prone to *unintentionally* encouraging overly flat loss functions, see Appendix B.1. Generally, flat regions in the loss function are very detrimental to training as uniform loss is given to non-uniform errors. Removal of the Softplus activation in the output can partially resolve this flatness issue; however, without it, the learned loss functions would violate the typical constraint that a loss function should be at least $\mathcal{C}^1$, *i.e.*, continuous in the zeroth and first derivatives.

Alternative smooth activations, such as Sigmoid, TanH, ELU, etc., can be used instead; however, due to their range-bounded limits, they are also prone to encouraging loss functions that have large flat regions when their activations saturate. Therefore, to inhibit this behavior, the unbounded leaky ReLU (Maas et al., 2013) is combined with the smooth ReLU, *i.e.*, SoftPlus (Dugas et al., 2000)

$$\varphi_{hidden}(x) = \frac{1}{\beta} \log(e^{\beta x} + 1) \cdot (1 - \gamma) + \gamma x. \tag{4}$$

This *smooth leaky ReLU* activation function with leak parameter $\gamma$ and smoothness parameter $\beta$ has desirable characteristics for representing a loss function. It is smooth and has linear asymptotic behavior necessary for tasks such as regression, where extrapolation of the learned loss function can often occur. Furthermore, as its output is not bounded when $\gamma > 0$, it does not encourage flatness in the learned loss function. See Appendix B.2 and D.2 for more details.

## 2.3 Loss Function Initialization

One challenge for online loss function learning is achieving a stable and performant initial set of parameters for the learned loss function. If $\phi$ is initialized poorly, too much time is spent on fixing $\phi$ in the early stages of the learning process, resulting in poor base convergence, or in the worst case, $f_{\theta}$ to diverge. To address this, offline loss function learning using *Meta-learning via Learned Loss* (ML$^3$) (Bechtle et al., 2021) is utilized to fine-tune the initial loss function to the base model prior to online loss function learning. The initialization process is summarized in Algorithm 1, where $\mathcal{S}_{init} = 2500$. In AdaLFL's initialization process one step on $\theta$ is taken for each step in $\phi$, *i.e.*, inner gradient steps $\mathcal{S}_{inner} = 1$. However, if $\mathcal{S}_{inner} > 1$, implicit differentiation (Lorraine et al., 2020; Gao et al., 2022) can instead be utilized to reduce the initialization process's memory and computational overhead.

---

**Algorithm 2** Loss Function Adaptation (Online)

---

**Input:** $\mathcal{M}_\phi \leftarrow$ Learned loss function (base-objective)
**Input:** $\mathcal{L}_\mathcal{T} \leftarrow$ Task loss function (meta-objective)

---

1: $\theta_0 \leftarrow$ Initialize parameters of base learner
2: **for** $t \in \{0, ..., \mathcal{S}_{train} - 1\}$ **do**
3:     $X, y \leftarrow$ Sample from $\mathcal{D}_{train}$
4:     $\mathcal{M}_{learned} \leftarrow \mathcal{M}_{\phi_t}(y, f_{\theta_t}(X))$
5:     $\theta_{t+1} \leftarrow \theta_t - \alpha \nabla_{\theta_i} \mathcal{M}_{learned}$
6:     $X, y \leftarrow$ Sample from $\mathcal{D}_{valid}$
7:     $\mathcal{L}_{task} \leftarrow \mathcal{L}_\mathcal{T}(y, f_{\theta_{t+1}}(X))$
8:     $\phi_{t+1} \leftarrow \phi_t - \eta \nabla_{\phi_t} \mathcal{L}_{task}$
9: **end for**

---

## 2.4 Online Meta-Optimization

To optimize $\phi$, unrolled differentiation is utilized in the outer loop to update the learned loss function after each update to the base model parameters $\theta$ in the inner loop, which occurs via vanilla backpropagation. This is conceptually the simplest way to optimize $\phi$ as all the intermediate iterates generated by the optimizer in the inner loop can be stored and then backpropagate through in the outer loop (Maclaurin et al., 2015). The full iterative learning process is summarized in Algorithm 2 and proceeds as follows: perform a forward pass $f_{\theta_t}(x)$ to obtain an initial set of predictions. The learned loss function $\mathcal{M}_\phi$ is then used to produce a base loss value

$$\mathcal{M}_{learned} = \mathcal{M}_{\phi_t}(y, f_{\theta_t}(x)). \tag{5}$$

Using $\mathcal{M}_{learned}$, the current weights $\theta_t$ are updated by taking a step in the opposite direction of the gradient of the loss with respect to the base model parameters $\theta_t$,

$$\begin{aligned} \theta_{t+1} &= \theta_t - \alpha \nabla_{\theta_t} \mathcal{M}_{\phi_t}(y, f_{\theta_t}(x)) \\ &= \theta_t - \alpha \nabla_{\theta_t} \mathbb{E}_{X,y}\big[\mathcal{M}_{\phi_t}(y, f_{\theta_t}(x))\big], \end{aligned} \tag{6}$$

where $\alpha$ is the base learning rate. This can be further decomposed via the chain rule,

$$\theta_{t+1} = \theta_t - \alpha \nabla_f \mathcal{M}_{\phi_t}(y, f_{\theta_t}(x)) \nabla_{\theta_t} f_{\theta_t}(x). \tag{7}$$

Importantly, all intermediate iterates generated by the base optimizer at the $t^{th}$ time-step when updating $\theta$ are stored in memory. The meta-parameters $\phi_t$ can now be updated to $\phi_{t+1}$ based on the learning progression made by $\theta$. Using $\theta_{t+1}$ as a function of $\phi_t$, compute a forward pass using the updated base weights $f_{\theta_{t+1}}(x)$ to obtain a new set of predictions. The instances can either be sampled from the training set or a held-out validation set. The new set of predictions is used to calculate the task loss $\mathcal{L}_\mathcal{T}$ to optimize $\phi_t$ through $\theta_{t+1}$,

$$\mathcal{L}_{task} = \mathcal{L}_\mathcal{T}(y, f_{\theta_{t+1}}(x)), \tag{8}$$

where $\mathcal{L}_\mathcal{T}$ is selected based on the respective application. For example, the squared error loss for the task of regression or the cross-entropy loss for classification. The task loss is a crucial component for embedding the end goal task into the learned loss function. Optimization of the current meta-loss network loss weights $\phi_t$ now occurs by taking the gradient of the task loss,

$$\begin{aligned} \phi_{t+1} &= \phi_t - \eta \nabla_{\phi_t} \mathcal{L}_\mathcal{T}(y, f_{\theta_{t+1}}(x)) \\ &= \phi_t - \eta \nabla_{\phi_t} \mathbb{E}_{X,y}\big[\mathcal{L}_\mathcal{T}(y, f_{\theta_{t+1}}(x))\big], \end{aligned} \tag{9}$$

where $\eta$ is the meta learning rate. The gradient computation is decomposed by applying the chain rule as shown in Equation (11) where the gradient with respect to the meta-loss network weights $\phi_t$ requires the updated model parameters $\theta_{t+1}$ from Equation (6).

$$\phi_{t+1} = \phi_t - \eta \nabla_f \mathcal{L}_\mathcal{T} \nabla_{\theta_{t+1}} f_{\theta_{t+1}} \nabla_{\phi_t} \theta_{t+1}(\phi_t) \tag{10}$$

$$= \phi_t - \eta \nabla_f \mathcal{L}_\mathcal{T} \nabla_{\theta_{t+1}} f_{\theta_{t+1}} \nabla_{\phi_t} [\theta_t - \alpha \nabla_{\theta_t} \mathcal{M}_{\phi_t}] \tag{11}$$

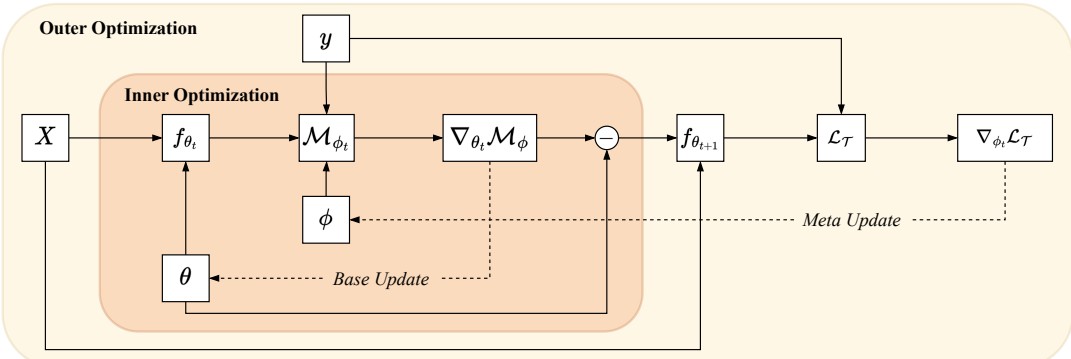

Figure 2: Computational graph of AdaLFL, where $\theta$ is updated using $\mathcal{M}_\phi$ in the inner loop (*Base Update*). The optimization path is tracked in the computational graph and then used to update $\phi$ based on the meta-objective in the outer loop (*Meta Update*). The dashed lines show the gradients for $\theta$ and $\phi$ with respect to their given objectives.

This process is repeated for a fixed number of gradient steps $S_{train}$, which is identical to what would typically be used for training $f_\theta$. An overview and summary of the full associated data flow between the inner and outer optimization of $\theta$ and $\phi$, respectively, is given in Figure 2.

## 3  Related Work

The method that we propose in this paper addresses the general problem of meta-learning a (base) loss function, *i.e.* loss function learning. Existing loss function learning methods can be categorized along two key axes, loss function representation, and meta-optimization. Frequently used representations in loss function learning include parametric (Gonzalez & Miikkulainen, 2020b; Raymond et al., 2023) and nonparametric (Liu et al., 2020; Li et al., 2022) genetic programming expression trees. In addition to this, alternative representations such as truncated Taylor polynomials (Gonzalez & Miikkulainen, 2021; Gao et al., 2021; 2022) and small feed-forward neural networks (Bechtle et al., 2021) has also been recently explored. Regarding meta-optimization, loss function learning methods have heavily utilized computationally expensive evolution-based methods such as evolutionary algorithms (Koza, 1994) and evolutionary strategies (Hansen & Ostermeier, 2001). While more recent approaches have made use of gradient-based approaches unrolled differentiation (Maclaurin et al., 2015), and implicit differentiation (Lorraine et al., 2020).

A common trait among these methods is that, in contrast to AdaLFL, they perform *offline* loss function learning, resulting in a severe short-horizon bias towards loss functions which are performant in the early stages of training but often have sub-optimal performance at the end of training. This short-horizon bias arises from how the various approaches compute their respective meta-objectives. In offline evolution-based approaches, the fitness, *i.e.*, meta-objective, is calculated by computing the performance at the end of a partial training session, *e.g.*, $\leq 1000$ gradient steps (Gonzalez & Miikkulainen, 2021; Raymond et al., 2023). A truncated number of gradient steps are required to be used as evolution-based methods evaluate the performance of a large number of candidate solutions, typically $L$ loss function over $K$ iterations where $25 \leq L, K \leq 100$. Therefore, performing full training sessions, which can be hundreds, thousands, or even millions of gradient steps for each candidate solution, is infeasible.

Regarding the existing gradient-based approaches, offline unrolled optimization requires the whole optimization path to be stored in memory; in practice, this significantly restricts the number of inner gradient steps before computing the meta-objective to only a small number of steps. Methods such as implicit differentiation can obviate these memory issues; however, it would still require a full training session in the inner loop, which is a prohibitive number of forward passes to perform in tractable time. Furthermore, the dependence of the model parameters on the meta-parameters increasingly shrinks and eventually vanishes as the number of steps increases (Rajeswaran et al., 2019).

### 3.1 Online vs Offline Loss Function Learning

The key algorithmic difference of AdaLFL from prior offline gradient-based methods (Bechtle et al., 2021; Gao et al., 2022) is that $\phi$ is updated after each update to $\theta$ in lockstep in a single phase as opposed to learning $\theta$ and $\phi$ in separate phases. This is achieved by not resetting $\theta$ after each update to $\phi$ (Algorithm 1, line 3), and consequently, $\phi$ has to adapt to each newly updated timestep such that $\phi = (\phi_0, \phi_1, \ldots, \phi_{S_{train}})$. In offline loss function learning, $\phi$ is learned separately at meta-training time and then is fixed for the full duration of the meta-testing phase where $\theta$ is learned and $\phi = (\phi_0)$. Another crucial difference is that in online loss function learning, there is an implicit meta-learning of the learning rate schedule and a built in early stopping mechanism, further discussed in Section 5.5.

## 4 Experimental Evaluation

In this section, the experimental setup for evaluating AdaLFL is presented. In summary experiments are conducted across seven open-access datasets and multiple well-established network architectures. The performance of AdaLFL is assessed against three benchmark methods. All experiments are implemented in `PyTorch` (Paszke et al., 2017) and `Higher` (Grefenstette et al., 2019), and the code is available at [1]. Further experimental details and results, including ablations on the loss function representation, are provided in Appendix C and D, respectively.

### 4.1 Benchmark Methods

AdaLFL is compared to three benchmark methods. The first is a baseline, which uses conventional hand-crafted losses, namely the mean squared error and cross-entropy loss. Next, is ML[3] (Bechtle et al., 2021) the offline counterpart of AdaLFL which meta-learns the loss function offline and does not adapt it during training. Finally, MetaLR, an equivalent algorithm for meta-learning a single scalar adaptive learning rate, which adjusts the learning rate online during training identical to AdaLFL, see Appendix C.1 for more details. Importantly, the choice of hyper-parameters between MetaLR, ML[3] and AdaLFL has been standardized to enable as fair of a comparison as possible.

### 4.2 Benchmark Tasks

Following the established literature on loss function learning, the regression datasets Communities and Crime (Redmond, 2009), Diabetes (Efron et al., 2004), and California Housing (Pace & Barry, 1997) are used as a simple domain to illustrate the capabilities of the proposed method. Following this classification datasets MNIST (LeCun et al., 1998), CIFAR-10, CIFAR-100 (Krizhevsky & Hinton, 2009), and SVHN (Netzer et al., 2011), are employed to assess the performance of AdaLFL to determine whether the results can generalize to larger, more challenging tasks. The original training-testing partitioning is used for all datasets, with 10% of the training instances allocated for validation. In addition, standard data augmentation techniques consisting of normalization, random horizontal flips, and cropping are applied to the training data of CIFAR-10, CIFAR-100, and SVHN during meta and base training.

### 4.3 Benchmark Models

A diverse set of well-established benchmark architectures are utilized to evaluate the performance of AdaLFL. For Communities and Crime, Diabetes, and California Housing a two hidden layer multi-layer perceptron (MLP) taken from (Baydin et al., 2018) is used. For MNIST, logistic regression (McCullagh et al., 1989), the previously mentioned MLP and the LeNet-5 (LeCun et al., 1998) architecture is used. Following this experiments are conducted on CIFAR-10, VGG-16 (Simonyan & Zisserman, 2015), AllCNN-C (Springenberg et al., 2014), ResNet-18 (He et al., 2016), and SqueezeNet (Iandola et al., 2016) are used. For the remaining datasets, CIFAR-100 and SVHN, WideResNet 28-10 and WideResNet 16-8 (Zagoruyko & Komodakis, 2016) are employed.

---

[1]GitHub Repository: `https://github.com/Decadz/Online-Loss-Function-Learning`

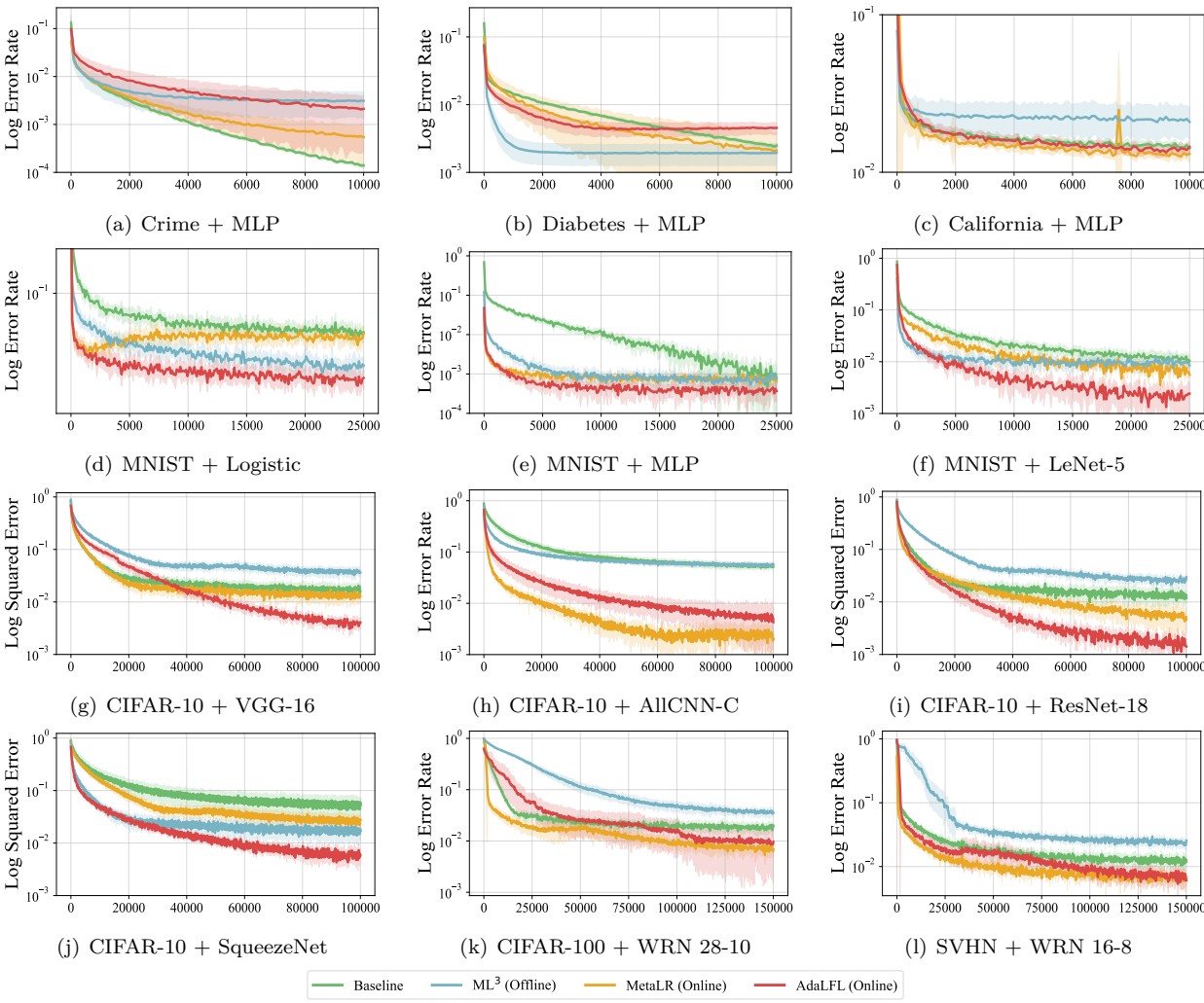

Figure 3: Mean learning curves across 10 independent executions of each algorithm on each task + model pair, showing the log of the training mean squared error or error rate (y-axis) against gradient steps (x-axis). Best viewed in color.

## 5    Results and Analysis

The results in Figure 3 show the average (log) training learning curves of AdaLFL compared with the baseline, MetaLR, and $ML^3$, across 10 executions of each dataset–model pair. Overall, AdaLFL demonstrates consistent improvements in convergence speed compared to the baseline and $ML^3$. These gains are evident across a wide range of tasks and architectures, with the exception of the regression datasets, which is due to regularization behavior as shown in Appendix D.4. Importantly, the errors obtained by AdaLFL at the end of training are typically lower than those of all compared methods, indicating that AdaLFL delivers both faster convergence and stronger overall performance.

On the more challenging tasks of CIFAR-10, CIFAR-100, and SVHN, AdaLFL improves noticeably over the baseline and $ML^3$. Improved performance on these datasets is achieved through AdaLFL's adaptive updating of the learned loss function throughout the learning process, which adapts to changes in the training dynamics. This is in contrast to $ML^3$, where the loss function remains static, resulting in poor performance on tasks where the training dynamics at the beginning of training vary significantly from those at the end of training.

Table 1: Results reporting the mean ± standard deviation of final inference testing mean squared error or error rate across 10 independent executions of each algorithm on each task + model pair (using no base learning rate scheduler).

| Task | Model | Baseline | $ML^3$ (Offline) | Meta-LR (Online) | AdaLFL (Online) |
|---|---|---|---|---|---|
| **Crime** | MLP [1] | 0.0274±0.0017 | 0.0270±0.0025 | 0.0274±0.0018 | **0.0263±0.0023** |
| **Diabetes** | MLP [1] | 0.0432±0.0013 | 0.0430±0.0012 | 0.0463±0.0013 | **0.0420±0.0014** |
| **California** | MLP [1] | 0.0157±0.0001 | 0.0276±0.0058 | 0.0154±0.0004 | **0.0151±0.0007** |
| **MNIST** | Logistic [2] | 0.0766±0.0009 | 0.0710±0.0010 | 0.0756±0.0008 | **0.0697±0.0010** |
| | MLP [1] | 0.0203±0.0006 | 0.0185±0.0004 | 0.0192±0.0007 | **0.0184±0.0006** |
| | LeNet-5 [3] | 0.0125±0.0007 | 0.0094±0.0005 | 0.0097±0.0013 | **0.0091±0.0004** |
| **CIFAR-10** | VGG-16 [4] | 0.1036±0.0049 | 0.1024±0.0055 | 0.0966±0.0087 | **0.0903±0.0032** |
| | AllCNN-C [5] | 0.1030±0.0062 | 0.1015±0.0055 | **0.0672±0.0068** | 0.0835±0.0050 |
| | ResNet-18 [6] | 0.0871±0.0057 | 0.0883±0.0041 | 0.0866±0.0056 | **0.0788±0.0035** |
| | SqueezeNet [7] | 0.1226±0.0080 | 0.1162±0.0052 | 0.1173±0.0065 | **0.1083±0.0049** |
| **CIFAR-100** | WRN 28-10 [8] | 0.3046±0.0087 | 0.3108±0.0075 | **0.2288±0.0019** | 0.2668±0.0283 |
| **SVHN** | WRN 16-8 [8] | 0.0512±0.0043 | 0.0500±0.0034 | **0.0367±0.0007** | 0.0441±0.0014 |

[1] (Baydin et al., 2018)  [2] (McCullagh et al., 1989)  [3] (LeCun et al., 1998)  [4] (Simonyan & Zisserman, 2015)
[5] (Springenberg et al., 2014)  [6] (He et al., 2016)  [7] (Iandola et al., 2016)  [8] (Zagoruyko & Komodakis, 2016)

Compared to MetaLR, AdaLFL generally achieves superior performance across tasks. This suggests that meta-learned loss functions capture nuanced meta-information about the training process that cannot be fully represented by meta-learning only the base learning rate, consistent with findings in (Raymond et al., 2023). Notably, on a few specific datasets: CIFAR-10 AllCNN-C, CIFAR-100 WRN28-10, and SVHN WRN16-8, MetaLR achieves slightly better performance. We hypothesize that this is due to the fixed learning rate settings adopted from prior works (Gonzalez & Miikkulainen, 2021; Raymond et al., 2023) being sub-optimal in these cases, making explicit meta-learning of the learning rate more effective.

## 5.1 Final Inference Testing Performance

Table 1 reporting the average mean squared error or error rate across 10 independent executions of each method. Across all tested problems, AdaLFL consistently outperform the baseline, these improvements are both substantial and stable, as reflected by the relatively small standard deviations across runs, indicating lower variability. Even though the models were originally designed and optimized around baseline loss functions, AdaLFL demonstrates clear and consistent gains across tasks. This result highlights the gains of using an adaptive meta-learned loss function compared to the typically used handcrafted loss functions.

Compared to $ML^3$, AdaLFL achieves better performance on all datasets. The improvements on MNIST are relatively modest; this suggests that the training dynamics on MNIST at the beginning of training are similar to those at the end; hence there are minimal gains from adapting the loss function online. While on more challenging datasets such as CIFAR-10, CIFAR-100, and SVHN, the gains are considerably larger, demonstrating the scalability and robustness of the approach. Regarding MetaLR, AdaLFL achieves better performance on most tasks, with the exception of CIFAR-10 AllCNN-C, CIFAR-100, and SVHN. This is likely due to the sub-optimal learning rates used on these tasks. Notably, MetaLR is also shown to outperform $ML^3$, this further highlights the importance of adaptation on large complex learning tasks.

The results attained by AdaLFL are promising given that the models tested were designed and optimized around the baseline loss functions. Larger performance gains may be attained using models designed specifically around meta-learned loss function (Kim et al., 2018; Elsken et al., 2020; Ding et al., 2022). Thus future work will explore learning the loss function in tandem with the network architecture.

Table 2: Average run-time of the entire learning process (end-to-end) for each benchmark method. Each algorithm is run on a single Nvidia RTX A5000, and results are reported in hours.

| Task and Model | Baseline | $ML^3$ | MetaLR | AdaLFL |
|---|---|---|---|---|
| Crime + MLP | 0.01 | 0.03 | 0.05 | 0.05 |
| Diabetes + MLP | 0.01 | 0.03 | 0.05 | 0.05 |
| California + MLP | 0.01 | 0.03 | 0.05 | 0.05 |
| MNIST + Logistic | 0.06 | 0.31 | 0.53 | 0.55 |
| MNIST + MLP | 0.06 | 0.32 | 0.53 | 0.56 |
| MNIST + LeNet-5 | 0.10 | 0.38 | 0.64 | 0.67 |
| CIFAR-10 + VGG-16 | 1.50 | 1.85 | 5.46 | 5.56 |
| CIFAR-10 + AllCNN-C | 1.41 | 1.72 | 5.39 | 5.53 |
| CIFAR-10 + ResNet-18 | 1.81 | 2.18 | 8.21 | 8.38 |
| CIFAR-10 + SqueezeNet | 1.72 | 2.02 | 7.84 | 7.88 |
| CIFAR-100 + WRN 28-10 | 8.81 | 10.3 | 48.71 | 50.49 |
| SVHN + WRN 16-8 | 7.32 | 7.61 | 23.33 | 24.75 |

## 5.2 Run-time Analysis

Table 2 reports the average run-time of all benchmark methods on all tasks, including the time to initialize the learning rate and loss function for MetaLR and AdaLFL, respectively. Notably, there are three key reasons why the computational overhead of AdaLFL is not as bad as it may at first seem. First, the baseline time excludes the implicit cost of manual hyper-parameter tuning for the loss function, initial learning rate, and learning rate schedule required to achieve reasonable performance (Goodfellow et al., 2016). Second, most the computational overhead of AdaLFL (and MetaLR) comes from storing the one-step optimization trajectory and backpropagating through it to update $\phi$. Importantly, this is an identical computation to those used in other meta-learning paradigms (Andrychowicz et al., 2016; Finn et al., 2017). Consequently, when combined with optimization-based meta-learning methods the computational overhead would be amortized, since the intermediate iterates generated by those algorithms can be reused (Li et al., 2017; Park & Oliva, 2019; Flennerhag et al., 2020; Baik et al., 2020; 2021). Finally, we take one meta-gradient step on $\phi$ per base step on $\theta$, but meta steps could instead occur periodically, e.g., every 100–1000 steps, to reduce the computational and memory overhead. As shown in Figures 9-18, the meta-learned loss functions interpolate very smoothly between their initial and final states, and in many cases converges well before the base model has finished training.

## 5.3 Inner Gradient Steps

In $ML^3$, (Bechtle et al., 2021) suggested taking only one inner step, i.e., setting $\mathcal{S}_{inner} = 1$ in Algorithm 1. A reasonable question to ask is whether increasing the number of inner steps to extend the horizon of the meta-objective past the first step will reduce the disparity in performance between $ML^3$ and AdaLFL. To answer this question, experiments are performed on CIFAR-10 AllCNN-C with $ML^3$ setting $\mathcal{S}_{inner} = \{1, 5, 10, 15, 20\}$. The results reported in Table 3 show that increasing the number of inner steps in $ML^3$ up to the limit of what is feasible in memory on a consumer GPU does *not* resolve the

Table 3: Results reporting the mean $\pm$ standard deviation of testing error rates when using an increasing number of inner gradient steps with $ML^3$.

| Method | CIFAR-10 + AllCNN-C |
|---|---|
| $ML^3$ ($\mathcal{S}_{inner} = 1$) | 0.1015±0.0055 |
| $ML^3$ ($\mathcal{S}_{inner} = 5$) | 0.0978±0.0052 |
| $ML^3$ ($\mathcal{S}_{inner} = 10$) | 0.0985±0.0050 |
| $ML^3$ ($\mathcal{S}_{inner} = 15$) | 0.0989±0.0049 |
| $ML^3$ ($\mathcal{S}_{inner} = 20$) | 0.0974±0.0061 |
| AdaLFL (Online) | **0.0835±0.0050** |

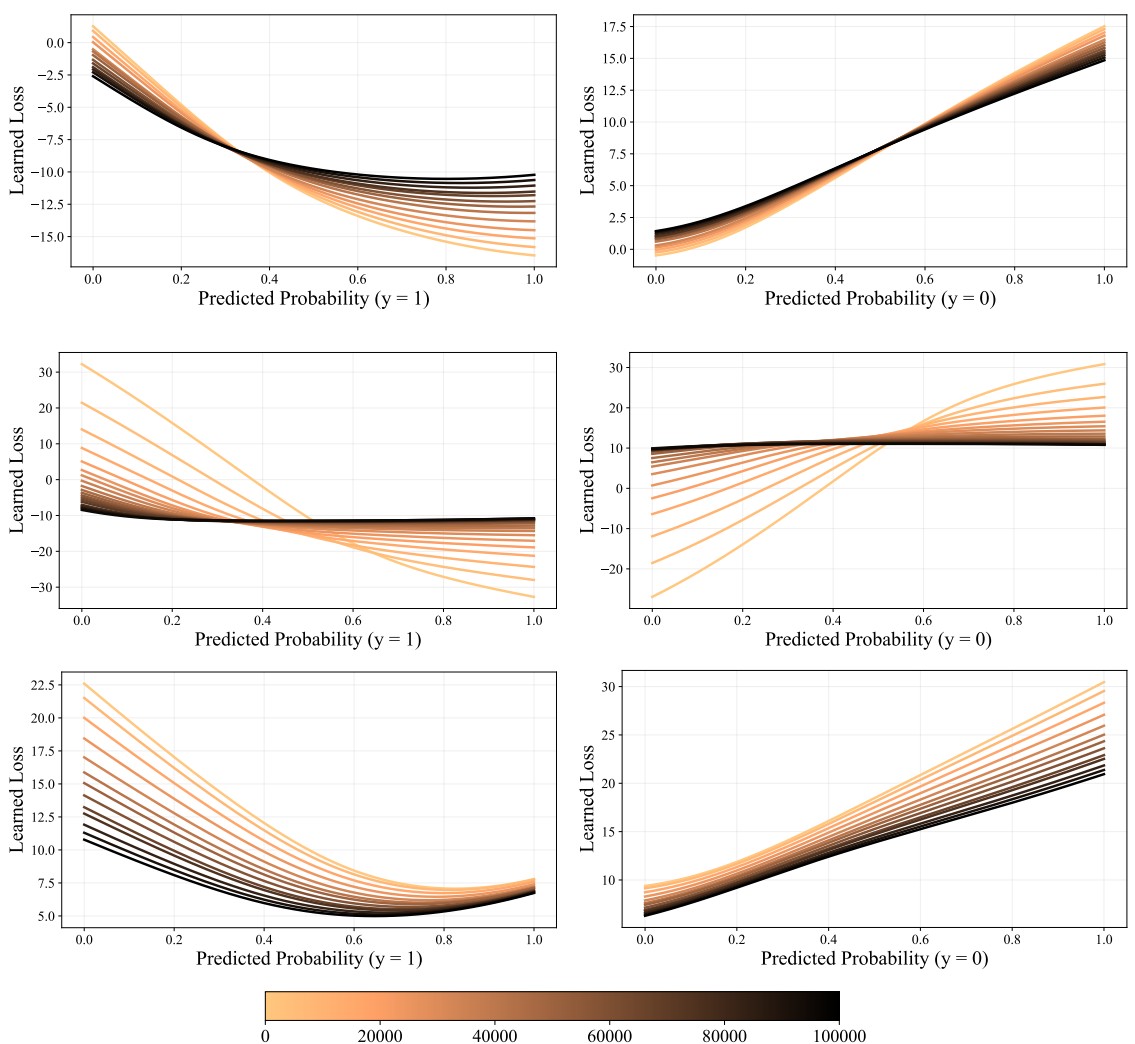

Figure 4: Loss functions generated by AdaLFL on the CIFAR-10 dataset, where each row represents a loss function and the color represents the current gradient step.

short horizon bias present in offline loss function learning. Furthermore, the results show that increasing the number of inner steps only results in marginal improvements in the performance over $\mathcal{S}_{inner} = 1$. Hence, offline learning methods that seek to obviate the memory issues of unrolled differentiation to allow for an increased number of inner steps, such as (Gao et al., 2022), which uses implicit differentiation, are still prone to a kind of short-horizon bias.

### 5.4 Visualizing Learned Loss Functions

To better understand why the meta-learned loss functions produced by AdaLFL are so performant, two of the learned loss functions are highlighted in Figure 4, where the learned loss function is plotted at equispaced intervals throughout the training. See Appendix D.6 for further examples of the diverse and creative loss function meta-learned by AdaLFL.

Analyzing the learned loss functions, it can be observed that the loss functions change significantly in their shape throughout the learning process. In both cases, the learned loss functions attributed strong penalties for severe misclassification at the start of the learning process, and then gradually pivoted to a more moderate or minor penalty as learning progressed. This behavior enables fast and efficient learning early on and reduces the sensitivity of the base model to outliers in the later stages of the learning process.

### 5.5 Implicit Tuning of Learning Rate Schedule

In offline loss function learning, it is known from (Gonzalez & Miikkulainen, 2021; Raymond et al., 2023) that there is implicit initial learning rate tuning of $\alpha$ when meta-learning a loss function since $\exists \alpha \exists \phi : \theta - \alpha \nabla_\theta \mathcal{L}_\mathcal{T} \approx \theta - \nabla_\theta \mathcal{M}_\phi$. Consequently, an emergent behavior, unique to online loss function learning, is that the adaptive loss function generated by AdaLFL implicitly embodies multiple different learning rates throughout the learning process hence often causing a fine-tuning of the fixed learning rate or of a predetermined learning rate schedule. Analyzing the learned loss functions in Figure 4, it can be observed that the scale of the learned loss function changes, confirming that implicit learning rate scheduling is occurring over time. In Appendix C.1 additional experiments are given comparing the performance of AdaLFL to a method for meta-learning the base learning rate.

### 5.6 Implicit Early Stopping Regularization

A unique property observed in the loss functions generated by AdaLFL is that often once base convergence is achieved the learned loss function will *intentionally* start to flatten or take on a parabolic form, as shown in Figure 4 (second row). This is implicitly a type of early stopping, also observed in related paradigms such as in hypergradient descent (Baydin et al., 2018), which meta-learns base learning rates. In hypergradient descent the learned learning rate has been observed to oscillate around 0 near the end of training, at times becoming negative, essentially terminating training. Implicit early stopping is beneficial as it is known to have a regularizing effect on model training (Yao et al., 2007); however, if not performed carefully it can also be detrimental to training due to terminating training prematurely. Therefore, in future work, we aim to further investigate regulating this behavior, as a potential avenue for further improving performance.

### 5.7 Implicit Label Smoothing Regularization

Another novel observation we make is that in many of the classification loss functions, such as in Figure 4 (third row), the target loss initially decreases as the model becomes more confident in its predictions. However, counterintuitively, as the predicted probability approaches 1 (*i.e.*, the model becomes very confident), the loss begins to increase. This behavior has been previously observed and theoretically studied in (Gonzalez & Miikkulainen, 2020a; Raymond et al., 2023), where it was shown to resemble a form of label smoothing regularization. This later inspired a new regularization method called sparse label smoothing regularization (Raymond, 2024). In such cases, the loss function penalizes overconfident predictions, thereby encouraging better generalization. Importantly, this is the first time this phenomenon has been observed in neural network parameterizations of the loss functions. Moreover, because AdaLFL learns the loss function adaptively, unlike prior methods that learn static loss functions, it implicitly meta-learns an *adaptive* form of label smoothing regularization which is dynamically adjusted throughout the learning process.

## 6 Conclusion

In this work, the first fully online approach to loss function learning has been proposed. The proposed technique, *Adaptive Loss Function Learning* (AdaLFL), infers the base loss function directly from the data and adaptively trains it with the base model parameters simultaneously using unrolled differentiation. The results showed that models trained with our method have enhanced convergence capabilities and inference performance compared with the *de facto* standard mean squared error and cross-entropy loss, and offline loss function learning method ML$^3$. Further analysis of the learned loss functions identified common patterns in the shape of the learned loss function, as well as revealed unique emergent behavior present only in adaptively learned loss functions. Namely, implicit tuning of the learning rate schedule, early stopping regularization, and an adaptive form of label smoothing regulization. While this work has solely set focus on meta-learning the loss function in isolation to better understand and analyze its properties, we believe that further benefits can be realized upon being combined with existing optimization-based meta-learning techniques.

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

# A   Extended Background

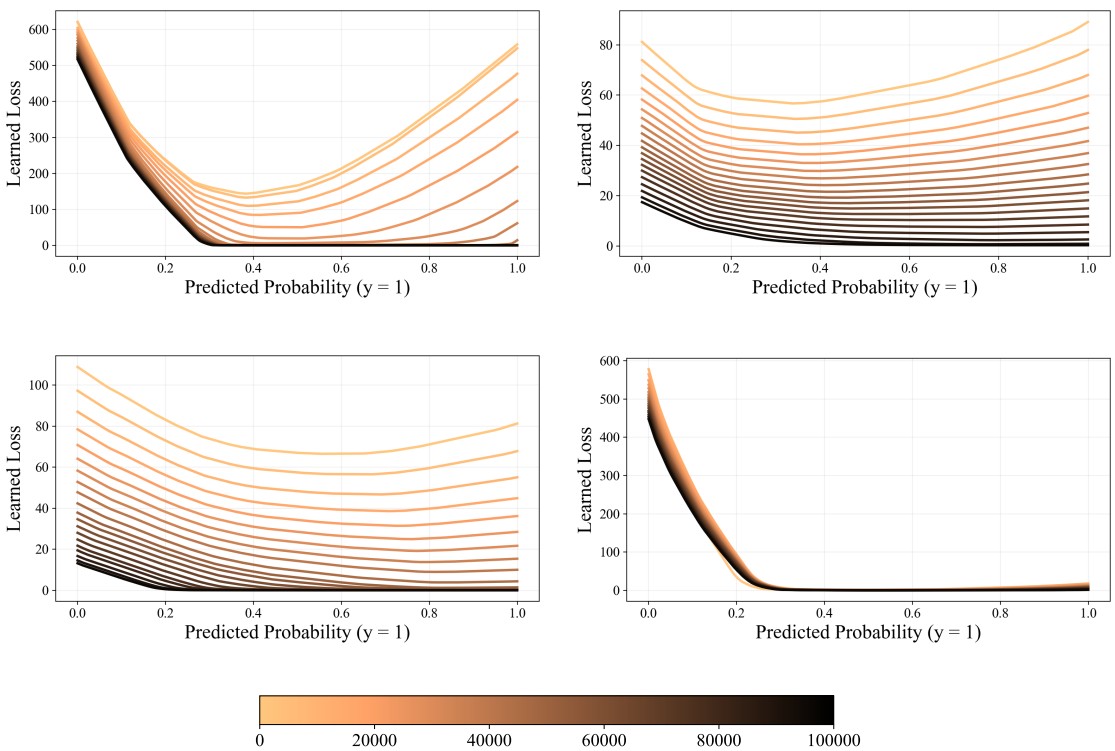

Figure 5: Four example loss functions generated by AdaLFL using the network architecture proposed in (Bechtle et al., 2021), which uses a softplus activation in the output layer, causing flattening behavior degrading learning performance.

## A.1   Online vs Offline Loss Function Learning

The key algorithmic difference of AdaLFL from prior offline gradient-based methods (Bechtle et al., 2021; Gao et al., 2022) is that $\phi$ is updated after each update to $\theta$ in lockstep in a single phase as opposed to learning $\theta$ and $\phi$ in separate phases. This is achieved by not resetting $\theta$ after each update to $\phi$ (Algorithm 1, line 3), and consequently, $\phi$ has to adapt to each newly updated timestep such that $\phi = (\phi_0, \phi_1, \ldots, \phi_{S_{train}})$. In offline loss function learning, $\phi$ is learned separately at meta-training time and then is fixed for the full duration of the meta-testing phase where $\theta$ is learned and $\phi = (\phi_0)$. Another crucial difference is that in online loss function learning, there is an implicit tuning of the learning rate schedule and implicit early stopping regularization, as discussed in Section 5.5 and 5.6, respectively.

## A.2   Alternative Paradigms

Although online loss function learning has not been explored in the meta-learning context, some existing research outside the subfield has previously explored the possibility of adaptive loss functions (Barron, 2019; Li et al., 2019; Wang et al., 2020). However, we emphasize that these approaches are categorically different in that they do not learn the loss function from scratch; instead, they interpolate between a small subset of handcrafted loss functions, updating the loss function after each epoch. Furthermore, in contrast to loss function learning which is both task and model-agnostic, these techniques are restricted to being task-specific, *e.g.*, face recognition only. Finally, this class of approaches does not implicitly tune the base learning rate $\alpha$, as is the case in loss function learning.

## B    Loss Function Representation

The representation of the learned loss function under consideration in AdaLFL is a simple feed-forward neural network. We consider the general case of a feed-forward neural network with one input layer, $L$ hidden layers, and one output layer. A hidden layer refers to a feed-forward mapping between two adjacent layers $h_{\phi^{(l)}}$ such that

$$h_{\phi^{(l)}}\left(h_{\phi^{(l-1)}}\right) = \varphi^{(l)}\left(\phi^{(l)\mathsf{T}} h_{\phi^{(l-1)}}\right), \forall l = 1, \ldots L, \tag{12}$$

where $\varphi(\cdot)^{(l)}$ refers to the element-wise activation function of the $l^{th}$ layer, and $\phi^{(l)}$ is the matrix of inter-connecting weights between $h_{\phi^{(l-1)}}$ and $h_{\phi^{(l)}}$. For the input layer, the mapping is defined as $h_{\phi^{(0)}}(y_i, f_\theta(x)_i)$, and for the output layer as $h_{\phi^{(out)}}(h_{\phi^{(L)}})$. Subsequently, the meta-learned loss function $\ell\phi$ parameterized by the set of meta-parameters $\phi = \{\phi_0, \ldots, \phi_l, \phi_{out}\}$ can be defined as a composition of feed-forward mappings such that

$$\ell\phi\left(y_i, f_\theta(x)_i\right) = h_{\phi^{(out)}}\left(h_{\phi^{(L)}}\left(\ldots\left(h_{\phi^{(0)}}\left(y_i, f_\theta(x)_i\right)\right)\ldots\right)\right) \tag{13}$$

which is applied output-wise across the $\mathcal{C}$ output channels of the ground truth and predicted labels, *e.g.*, applied to each index of the one-hot encoded class vector in classification, or to each continuous output in regression. The loss value produced by $\ell\phi$ is then summed across the output channel to reduce the loss vector into its final scalar form

$$\mathcal{M}_\phi(y, f_\theta(x)) = \frac{1}{\mathcal{C}} \sum_{i=0}^{\mathcal{C}} \ell_\phi(y_i, f_\theta(x)_i). \tag{14}$$

### B.1    Network Architecture

The learned loss function used in our experiments has $L = 2$ hidden layers and 40 hidden units in each layer, inspired by the network configuration utilized in *Meta-Learning via Learned Loss* (ML$^3$ Supervised) (Bechtle et al., 2021). We found no consistent improvement in performance across our experiments by increasing or decreasing the number of hidden layers or nodes. However, it was found that the choice of non-linear activations used in ML$^3$, was highly prone to encouraging poor-performing loss functions with large flat regions, as shown in Figure 5.

In ML$^3$, rectified linear units, $\varphi_{ReLU}(x) = max(0, x)$, are used in the hidden layers and the smooth SoftPlus $\varphi_{softplus} = \log(e^{\beta x} + 1)$ is used in the output layer to enforce the optional constraint that $\mathcal{M}_{learned}$ should be non-negative, *i.e.*, $\forall y \forall f_\theta(x) \mathcal{M}_{\phi_t}(y, f_\theta(x)) \geq 0$. An adverse side-effect of using the softplus activation in the output is that all negative inputs to the output layer go to 0, resulting in flat regions in the learned loss. Furthermore, removal of the output activation does not resolve this issue, as ReLU, as well as other common activations such as Sigmoid, TanH, and ELU, are also bounded and are prone to causing flatness when their activations saturate, a common occurrence when taking gradients through long unrolled optimization paths (Antoniou et al., 2019).

### B.2    Smooth Leaky ReLU

To inhibit the flattening behavior of learned loss functions, a range unbounded activation function should be used. A popular activation function that is unbounded (when the leak parameter $\gamma < 0$) is the *Leaky ReLU* (Maas et al., 2013)

$$\varphi_{leaky}(x) = \max(\gamma \cdot x, x) \tag{15}$$
$$= \max(0, x) \cdot (1 - \gamma) + \gamma x. \tag{16}$$

However, it is typically assumed that a loss function should be *at least* $\mathcal{C}^1$, *i.e.*, continuous in the zeroth and first derivatives. Fortunately, there is a smooth approximation to the ReLU, commonly referred to as the *SoftPlus* activation function (Dugas et al., 2000), where $\beta$ (typically set to 1) controls the smoothness.

$$\varphi_{smooth}(x) = \frac{1}{\beta} \cdot \log(e^{\beta x} + 1) \tag{17}$$

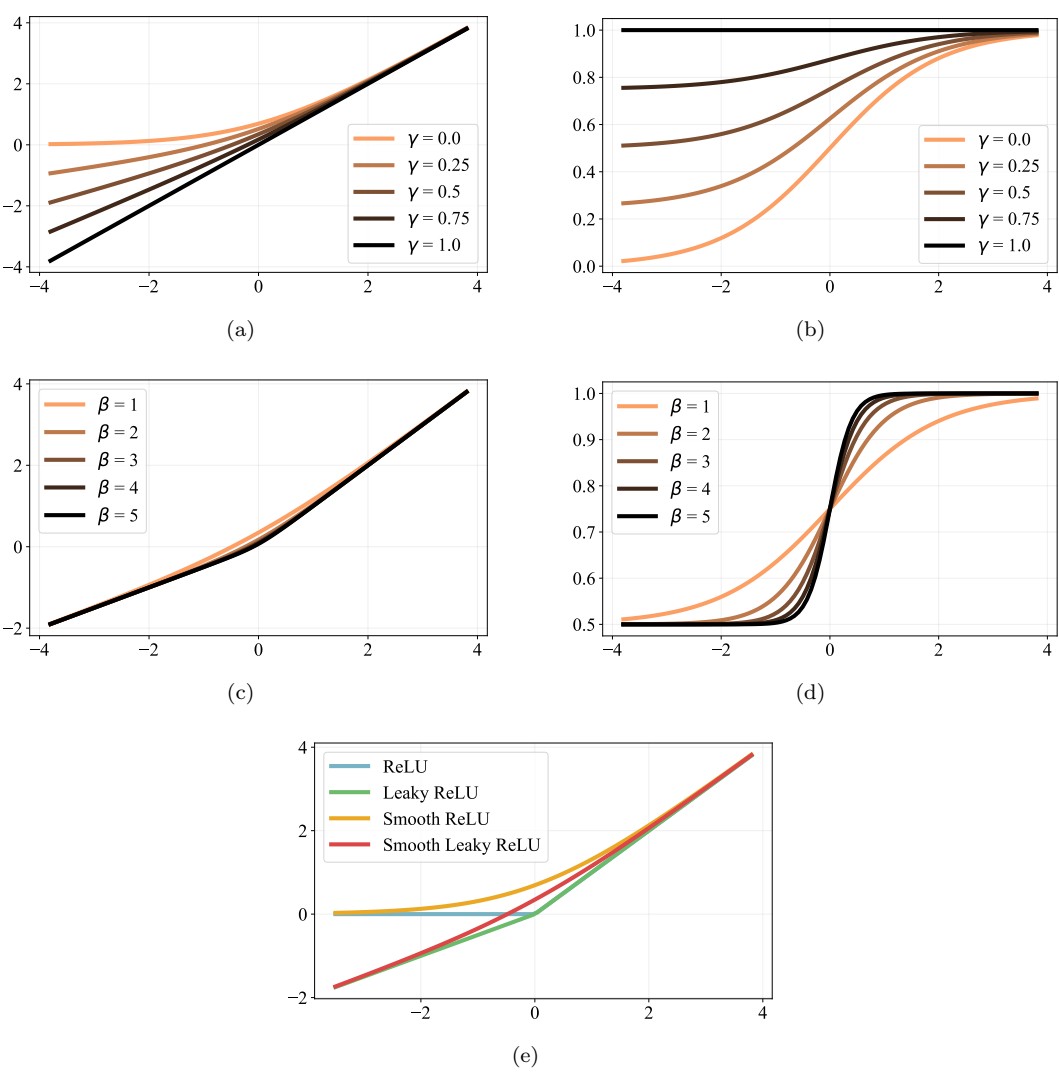

Figure 6: The proposed activation function and its corresponding derivatives when shifting $\gamma$ are shown in (a) and (b), respectively. In (c) and (d) the activation function and its derivatives when shifting $\beta$ are shown. Finally, in (c), the smooth leaky ReLU is contrasted with the original smooth and leaky variants ReLU.

The leaky ReLU is combined with the smooth ReLU by taking the term $max(0, x)$ from Equation (16) and substituting it with the smooth SoftPlus defined in Equation (17) to construct a smooth approximation to the leaky ReLU,

$$\varphi_{hidden}(x) = \frac{1}{\beta} \log(e^{\beta x} + 1) \cdot (1 - \gamma) + \gamma x, \tag{18}$$

where the derivative of the smooth leaky ReLU with respect to the input $x$ is

$$\varphi'_{hidden}(x) = \frac{d}{dx} \left[ \frac{\log(e^{\beta x} + 1) \cdot (1 - y)}{\beta} + \gamma x \right] \tag{19}$$

$$= \frac{\frac{d}{dx}[\log(e^{\beta x} + 1)] \cdot (1 - y)}{\beta} + \gamma \tag{20}$$

$$= \frac{\frac{d}{dx}[e^{\beta x} + 1] \cdot (1 - y)}{\beta \cdot e^{\beta x} + 1} + \gamma \tag{21}$$

$$= \frac{e^{\beta x} \cdot \beta \cdot (1 - y)}{\beta \cdot e^{\beta x} + 1} + \gamma \tag{22}$$

$$= \frac{e^{\beta x}(1 - \gamma)}{e^{\beta x} + 1} + \gamma \tag{23}$$

$$= \frac{e^{\beta x}(1 - \gamma)}{e^{\beta x} + 1} + \frac{\gamma(e^{\beta x} + 1)}{e^{\beta x} + 1} \tag{24}$$

$$= \frac{e^{\beta x} + \gamma}{e^{\beta x} + 1}. \tag{25}$$

The smooth leaky ReLU and its corresponding derivatives are shown in Figure 6. Early iterations of AdaLFL learned $\gamma$ and $\beta$ simultaneously with the network weights $\phi$, however; empirically, we found that setting $\gamma = 0.01$ and $\beta = 10$ gave adequate inference performance across our experiments.

## C   Experimental Setup

For all datasets, we use the original training–testing split, with 10% of the training data held out as a validation set. The validation set is used to optimize the training setup and tune hyper-parameters for the baseline models, ensuring they represent the strongest possible comparisons. Standard data augmentation techniques—normalization, random horizontal flips, and random cropping—are applied to the training data of CIFAR-10, CIFAR-100, and SVHN during both meta-training and base training.

In the inner loop, all regression models are trained using stochastic gradient descent (SGD) with a base learning rate of $\alpha = 0.001$. Classification models are trained with SGD using a base learning rate of $\alpha = 0.01$, and on CIFAR-10, CIFAR-100, and SVHN, Nesterov momentum 0.9 and weight decay 0.0005 are applied. These choices reflect the settings that gave the best performance on the validation set, while the remaining base-model hyper-parameters follow the standard configurations reported in the literature, consistent with the setup in (Gonzalez & Miikkulainen, 2021).

To initialize $\mathcal{M}_\phi$, $\mathcal{S}_{init} = 2500$ steps are taken in offline mode with a meta learning rate of $\eta = 1e - 3$. In contrast, in online mode, a meta learning rate of $\eta = 1e - 5$ is used (note, a high meta learning rate in online mode can cause a jittering effect in the loss function, which can cause training instability). For meta-optimization, the Adam optimizer (Kingma & Ba, 2015) is used in the outer loop for both initialization and online adaptation.

All experimental results reported show the average across 10 independent executions with different seeds to analyze algorithmic consistency. To ensure fairness, all methods are initialized with identical parameters under the same random seed, such that any differences in variance can be attributed to the respective algorithms and their loss functions. Furthermore, the choice of hyper-parameters between MetaLR, ML$^3$, and AdaLFL has been standardized to allow for a fair comparison.

### C.1 Meta-Learned Learning Rate

A notable finding presented in Section 5.5 of the main manuscript is the intimate relationship between learning a loss function and learning a learning rate/learning rate schedule, which is due to learned loss functions not just learning shape, but also learning scale. Given this relationship, it is interesting to compare and contrast the performance of AdaLFL to a method for meta-learning a base learning rate online. To construct a fair comparison, we use an identical learning setup to that used in AdaLFL, which allows us to control for the hyper-parameter settings. The algorithm, which we further refer to as *Meta-LR* is presented in Algorithms 3 and 4, uses an offline initialization process to find the best initial base learning rate, following which learning subsequently progresses to an online adaptation process.

---

**Algorithm 3** Learning Rate Initialization (Offline)

---

**Input:** $\mathcal{L}_{\mathcal{T}} \leftarrow$ Task loss function (meta-objective)

---

1: $\alpha_0 \leftarrow$ Initialize the base learning rate
2: **for** $t \in \{0, ..., \mathcal{S}_{init} - 1\}$ **do**
3:    $\theta_0 \leftarrow$ Reset parameters of base learner
4:    **for** $i \in \{0, ..., \mathcal{S}_{inner} - 1\}$ **do**
5:       $X, y \leftarrow$ Sample from $\mathcal{D}_{train}$
6:       $\theta_{i+1} \leftarrow \theta_i - \alpha_t \nabla_{\theta_i} \mathcal{L}_{\mathcal{T}}(y, f_{\theta_i}(X))$
7:    **end for**
8:    $X, y \leftarrow$ Sample from $\mathcal{D}_{valid}$
9:    $\mathcal{L}_{task} \leftarrow \mathcal{L}_{\mathcal{T}}(y, f_{\theta_{i+1}}(X))$
10:    $\alpha_{t+1} \leftarrow \alpha_t - \eta \nabla_{\alpha_t} \mathcal{L}_{task}$
11: **end for**

---

---

**Algorithm 4** Learning Rate Adaptation (Online)

---

**Input:** $\alpha \leftarrow$ Learned learning rate
**Input:** $\mathcal{L}_{\mathcal{T}} \leftarrow$ Task loss function

---

1: $\theta_0 \leftarrow$ Initialize parameters of base learner
2: **for** $t \in \{0, ..., \mathcal{S}_{train} - 1\}$ **do**
3:    $X, y \leftarrow$ Sample from $\mathcal{D}_{train}$
4:    $\theta_{t+1} \leftarrow \theta_t - \alpha_i \nabla_{\theta_t} \mathcal{L}_{\mathcal{T}}(y, f_{\theta_t}(X))$
5:    $X, y \leftarrow$ Sample from $\mathcal{D}_{valid}$
6:    $\mathcal{L}_{task} \leftarrow \mathcal{L}_{\mathcal{T}}(y, f_{\theta_{t+1}}(X))$
7:    $\alpha_{t+1} \leftarrow \alpha_t - \eta \nabla_{\alpha_t} \mathcal{L}_{task}$
8: **end for**

---

Table 4: Experimental results exploring alternative loss function representations based on Taylor polynomial parameterizations reporting the mean ± standard deviation of final inference testing mean squared error or error rate across 10 independent executions of each algorithm on each task + model pair.

| Task | Model | Quadratic-TP (Online) | Cubic-TP (Online) | AdaLFL (Online) |
|------|-------|----------------------|-------------------|-----------------|
| **Crime** | MLP | - | **0.0254±0.0015** | 0.0263±0.0023 |
| **Diabetes** | MLP | - | **0.0418±0.0041** | 0.0420±0.0014 |
| **California** | MLP | - | 0.0783±0.0167 | **0.0151±0.0007** |
| **MNIST** | Logistic | 0.0810±0.0281 | 0.0707±0.0009 | **0.0697±0.0010** |
| | MLP | 0.0205±0.0008 | 0.0185±0.0007 | **0.0184±0.0006** |
| | LeNet-5 | 0.1357±0.0728 | 0.0096±0.0006 | **0.0091±0.0004** |
| **CIFAR-10** | VGG-16 | 0.1442±0.0025 | 0.1439±0.0027 | **0.0903±0.0032** |
| | AllCNN-C | 0.1086±0.0100 | 0.0908±0.0020 | **0.0835±0.0050** |
| | ResNet-18 | 0.1133±0.0033 | 0.1309±0.0070 | **0.0788±0.0035** |
| | SqueezeNet | 0.1506±0.0092 | 0.1367±0.0041 | **0.1083±0.0049** |
| **CIFAR-100** | WRN 28-10 | 0.2952±0.0220 | 0.2934±0.0621 | **0.2668±0.0283** |
| **SVHN** | WRN 16-8 | 0.0494±0.0000 | 0.0431±0.0000 | **0.0441±0.0014** |

## D    Further Experiments

### D.1    Loss Function Representations

In AdaLFL a two-hidden-layer feedforward neural network is used for the loss function representation, this was inspired by its use in prior studies (Bechtle et al., 2021; Psaros et al., 2022). We chose this representation as it has more expressive power than both quadratic and cubic Taylor polynomials, which were used in (Gonzalez & Miikkulainen, 2021) and (Gao et al., 2021; 2022), respectively. Although the best representation for learned loss functions is not under investigation; it is important to note that the proposed method of online meta-optimization discussed in Section 2.4 makes no assumptions about the underlying representation used for the learned loss function. Therefore, alternative representations can be used in AdaLFL.

In Table 4, results comparing and contrasting the performance between different learned loss function representations are presented. Specifically, we contrast the performance of AdaLFL which uses a feed-forward neural network (NN) with smooth leaky ReLU activations against the aforementioned quadratic and cubic Taylor polynomials (TP) representation. The results show that the NN representation has on average the best performance and consistency in contrast to quadratic and cubic Taylor polynomials, with better performance and very little variance between independent executions on all datasets except the two small regression datasets Crime and Diabetes. These results demonstrate the superiority of the NN representation for learned loss functions, especially when dealing with relatively large learning tasks where expressive behavior is important. Note, that on the regression datasets, we found that the majority of the quadratic TP experiments diverged, even with hyper-parameter tuning.

### D.2    Loss Network Activation Function

An important difference between AdaLFL's neural network representation and prior neural network-based learned loss function representation such as the one used in $ML^3$, is the use of smooth leaky ReLU activation functions presented in Section 2.2 of the main manuscript. This new activation function resolves many issues with the prior network design; however, it remains to be seen how much of the performance improvement can be attributed to the newly proposed smooth leaky ReLU activation function vs the newly proposed online optimization algorithm.

Table 5: Experimental results ablating the newly proposed smooth leaky ReLU activation function, reporting the mean ± standard deviation of final inference testing mean squared error or error rate across 10 independent executions of each algorithm on each task + model pair.

| Task | Model | ML$^3$ ReLU (Offline) | ML$^3$ SLReLU (Offline) | AdaLFL (Online) |
|------|-------|----------------------|------------------------|-----------------|
| **Crime** | MLP | 0.0270±0.0025 | 0.0274±0.0029 | **0.0263±0.0023** |
| **Diabetes** | MLP | 0.0481±0.0020 | 0.0430±0.0012 | **0.0420±0.0014** |
| **California** | MLP | 0.0346±0.0087 | 0.0276±0.0058 | **0.0151±0.0007** |
| **MNIST** | Logistic | 0.0782±0.0117 | 0.0710±0.0015 | **0.0697±0.0010** |
| | MLP | 0.0167±0.0021 | 0.0185±0.0004 | **0.0184±0.0006** |
| | LeNet-5 | 0.0095±0.0006 | 0.0094±0.0005 | **0.0091±0.0004** |
| **CIFAR-10** | VGG-16 | 0.1034±0.0058 | 0.1024±0.0055 | **0.0903±0.0032** |
| | AllCNN-C | 0.1087±0.0174 | 0.1015±0.0055 | **0.0835±0.0050** |
| | ResNet-18 | 0.0972±0.0259 | 0.0883±0.0041 | **0.0788±0.0035** |
| | SqueezeNet | 0.1282±0.0086 | 0.1162±0.0052 | **0.1083±0.0049** |
| **CIFAR-100** | WRN 28-10 | 0.3114±0.0063 | 0.3108±0.0075 | **0.2668±0.0283** |
| **SVHN** | WRN 16-8 | 0.0500±0.0034 | 0.0502±0.0032 | **0.0441±0.0014** |

In Table 5, results are presented comparing and contrasting the performance between offline loss function learning (*i.e.* ML$^3$) with the standard ReLU + SoftPlus network architecture, and the new smooth leaky ReLU network architecture. The results show that on most tasks the new activation function improves performance compared to the conventional architecture used in ML$^3$. However, this performance improvement is not a significant contributing factor compared to the change in optimization algorithm, *i.e.* going from offline to online meta-learning.

### D.3 Second-Order Hyperparameter Sensitivity

We further investigated the sensitivity of AdaLFL to second-order hyperparameters, with particular focus on the meta-level learning rate $\eta$, which was the only parameter found to significantly affect performance, a behavior consistent with observations in other widely used meta-learning algorithms such as MAML (Finn et al., 2017). In the online adaptation setting, setting $\eta$ too high can cause the learned loss function to change too abruptly after each update, leading to unstable or oscillatory training dynamics. To evaluate this effect, we performed an ablation study on MNIST using LeNet-5 with varying values for the meta learning rate $\eta \in \{10^{-1}, \ldots, 10^{-6}\}$. The results presented in Table 6, demonstrate that AdaLFL maintains stable and consistent performance for all but the largest value of $\eta$, where a there is a slight degradation in accuracy. Overall, these findings indicate that AdaLFL is robust to the choice of $\eta$ within a relatively broad range, providing evidence of its stability with respect to second-order hyperparameter sensitivity.

Table 6: Ablating the second-order hyperparameter sensitivity to the meta-level learning rate $\eta$. Results report the mean ± standard deviation of testing error rate across 10 independent executions of each task+model.

| Task | Model | AdaLFL |
|------|-------|--------|
| **MNIST** | LeNet-5 ($\eta = 10^{-1}$) | 0.0191 ± 0.0070 |
| | LeNet-5 ($\eta = 10^{-2}$) | 0.0159 ± 0.0017 |
| | LeNet-5 ($\eta = 10^{-3}$) | 0.0104 ± 0.0011 |
| | LeNet-5 ($\eta = 10^{-4}$) | 0.0094 ± 0.0007 |
| | LeNet-5 ($\eta = 10^{-5}$) | 0.0091 ± 0.0004 |
| | LeNet-5 ($\eta = 10^{-6}$) | 0.0088 ± 0.0004 |

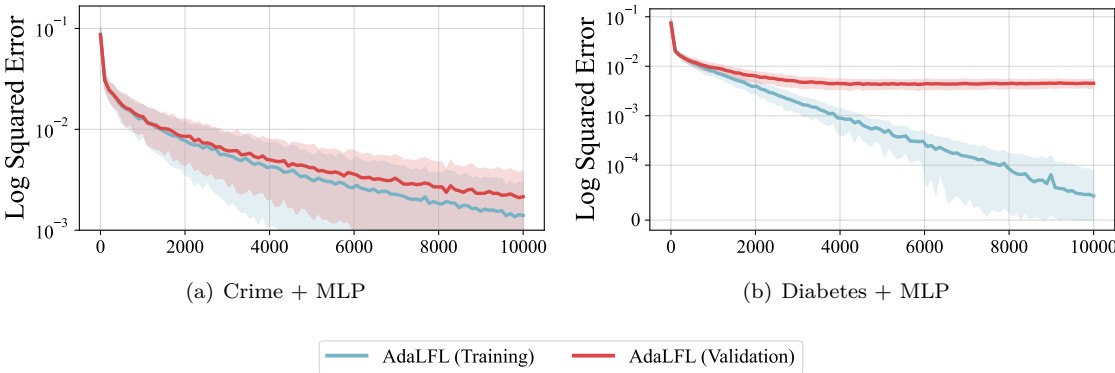

(a) Crime + MLP              (b) Diabetes + MLP

Figure 7: Mean learning curves across 10 independent executions of AdaLFL showing the training mean squared error (y-axis) against gradient steps (x-axis) when taking meta gradient steps on the meta-training (blue) vs meta-validation (red) set.

## D.4 Regularization and Meta-Objectives

When computing the meta-objective of AdaLFL (Equation 8 in the main manuscript) $\mathcal{L}_{task} = \mathcal{L}_{\mathcal{T}}(y, f_{\theta_{t+1}}(x))$, the instances can either be sampled from $\mathcal{D}_{train}$ or $\mathcal{D}_{valid}$. Consequently, the learned loss function can either optimize for in-sample performance or out-of-sample generalization, respectively. This behavior is shown on the Communities and Crime and Diabetes dataset, where as shown in Figure 7, optimizing the meta-objective using training samples

Table 7: Results reporting the mean ± standard deviation testing mean squared error across 10 independent executions of each algorithm on each task + model pair.

| Task | AdaLFL (Training) | AdaLFL (Validation) |
|------|-------------------|---------------------|
| Crime | 0.0267±0.0022 | **0.0265±0.0021** |
| Diabetes | 0.0468±0.0016 | **0.0420±0.0014** |

results in the training error quickly approaching 0. In contrast, when using the validation samples the training error does not converge as quickly and to as low of a training error value. This behavior is a form of regularization since as shown in Table 7 the final inference testing error is superior when using validation samples on both the Communities and Crime and Diabetes datasets. This is an important discovery as it suggests that loss function learning can induce a form of regularization, similar to the findings in (Gonzalez & Miikkulainen, 2021; 2020a; Raymond et al., 2023).

## D.5 Generalization Dynamics During Training

To further assess the generalization performance of AdaLFL during training, we conducted experiments on MNIST using the LeNet-5 base model. The dataset was partitioned into four subsets: training, validation, evaluation, and testing. The validation set was used to learn the loss function parameters $\phi$, while the evaluation set, a distinct held-out subset, was employed to monitor out-of-sample performance throughout training. As shown in Figure 8, the proposed method achieves faster convergence and improved generalization compared to the baseline. These results indicate that AdaLFL not only accelerates optimization but also enhances robustness against overfitting, leading to more stable learning dynamics.

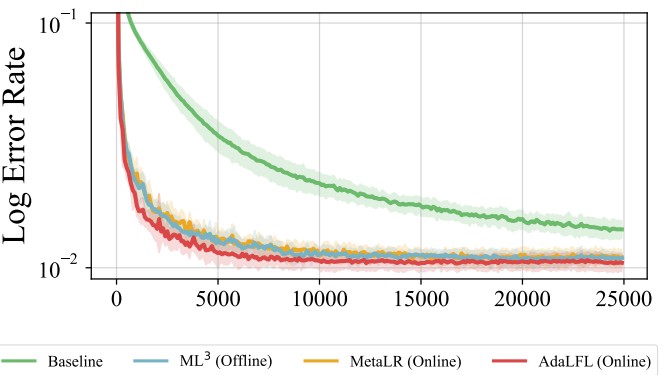

Figure 8: Mean leaning curves across 10 independent executions of each algorithm on MNIST+LeNet-5, showing the log of the validation error rate (y-axis) against gradient steps (x-axis).

## D.6 Learned Loss Functions (Extended)

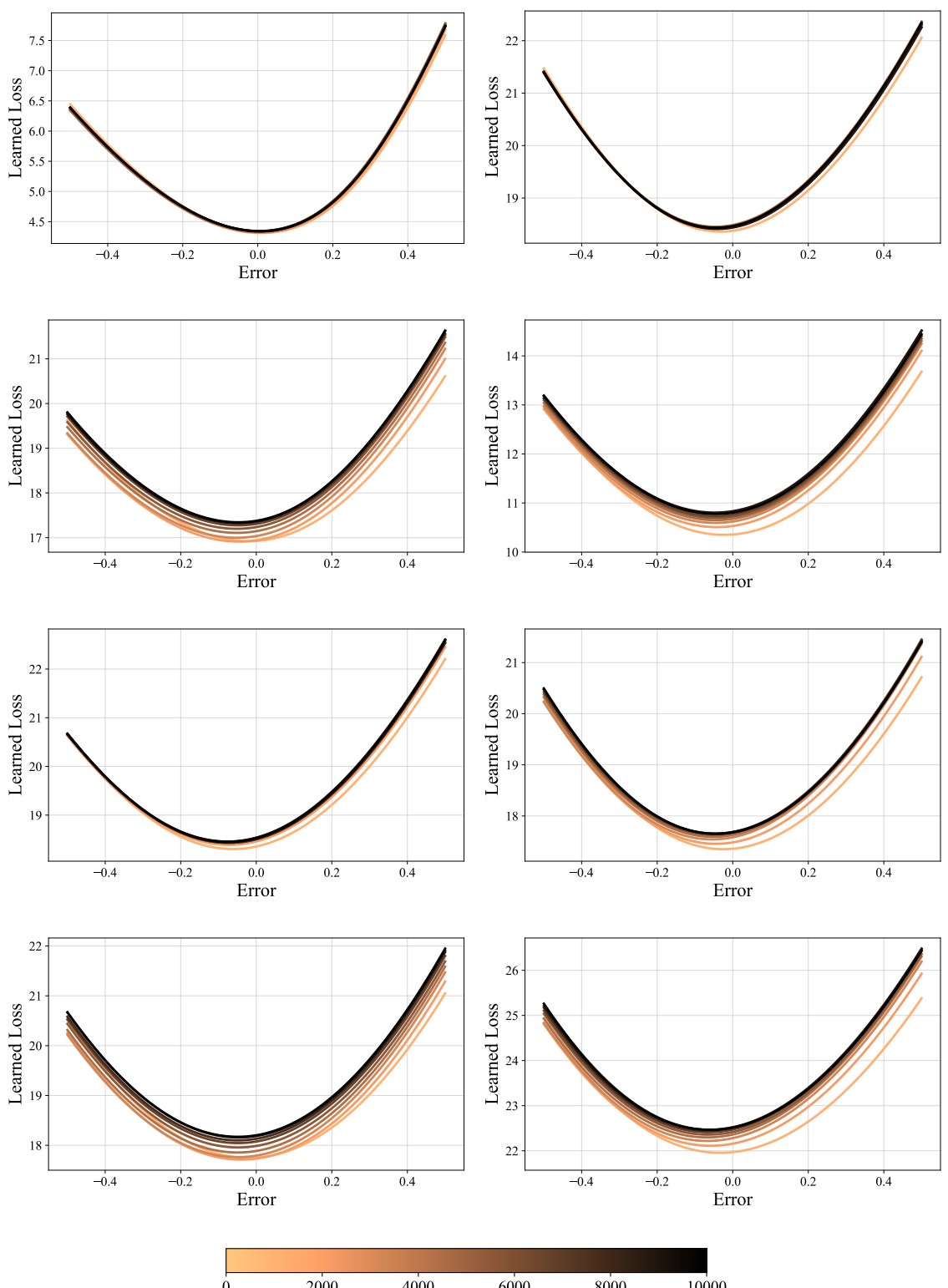

Figure 9: Loss functions generated by AdaLFL on the Communities and Crime dataset, where each plot represents a loss function, and the color represents the current gradient step.

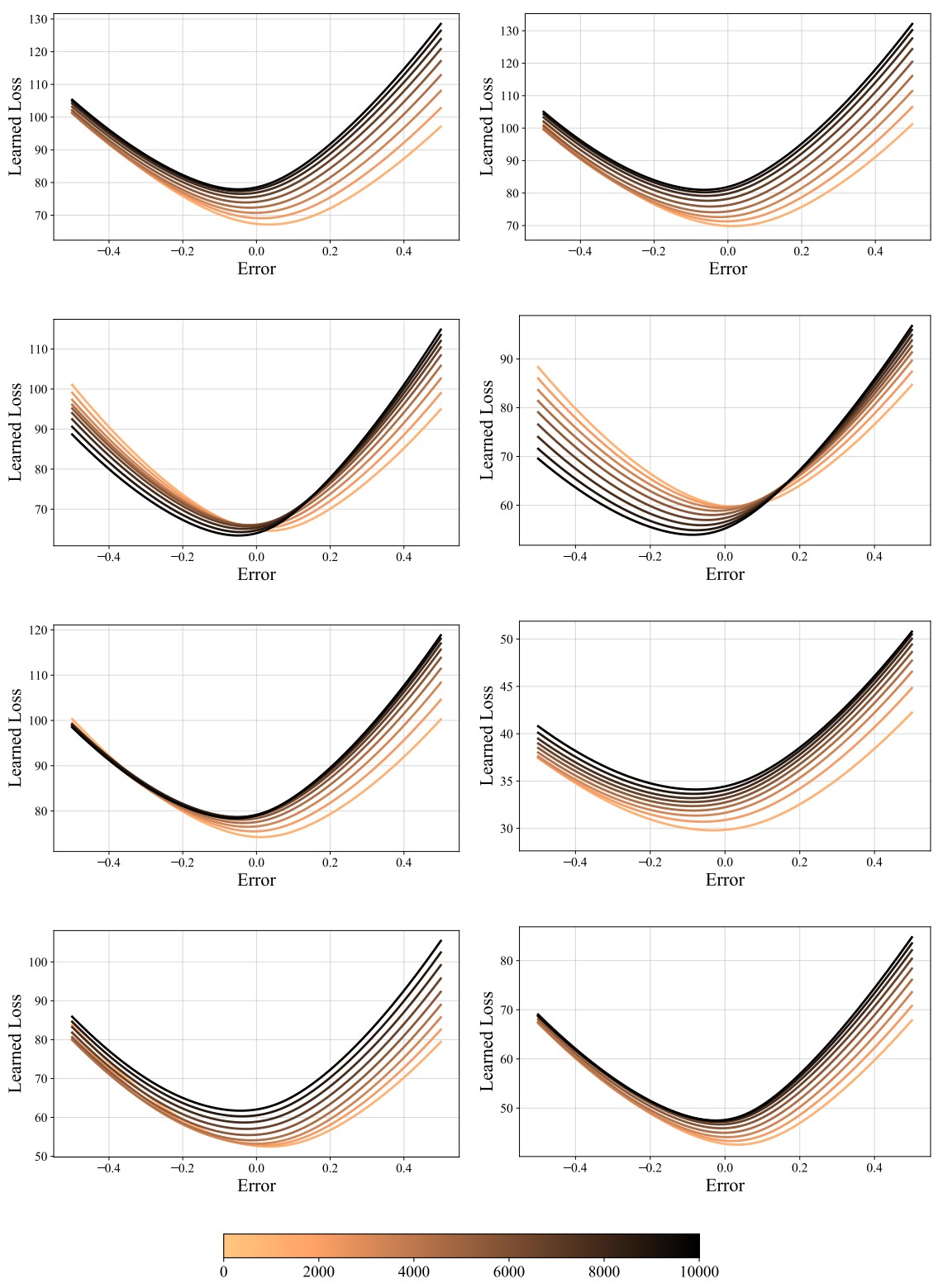

Figure 10: Loss functions generated by AdaLFL on the Diabetes dataset, where each plot represents a loss function, and the color represents the current gradient step.

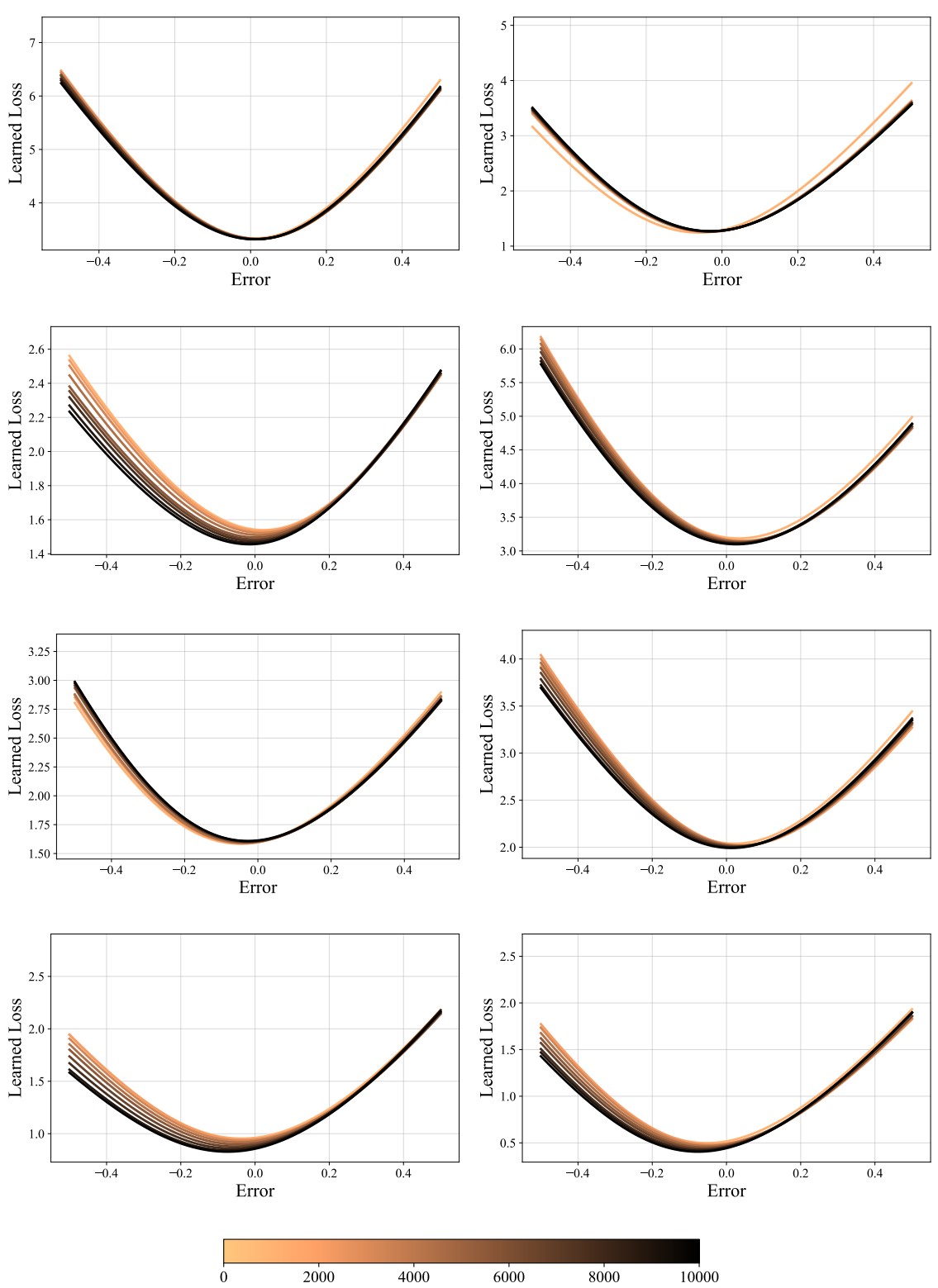

Figure 11: Loss functions generated by AdaLFL on the California Housing dataset, where each plot represents a loss function, and the color represents the current gradient step.

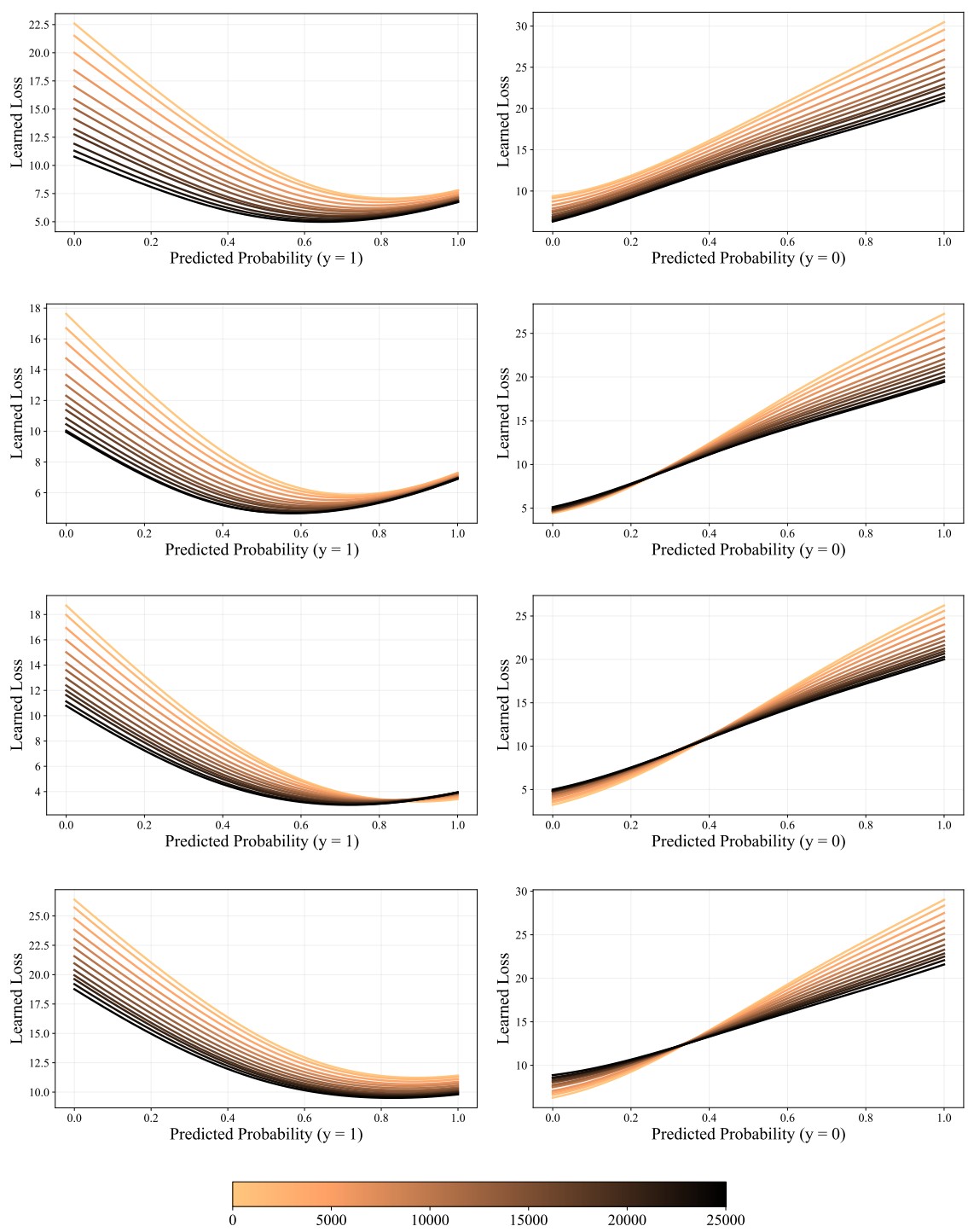

Figure 12: Loss functions generated by AdaLFL on the MNIST dataset, where each row represents a loss function, and the color represents the current gradient step.

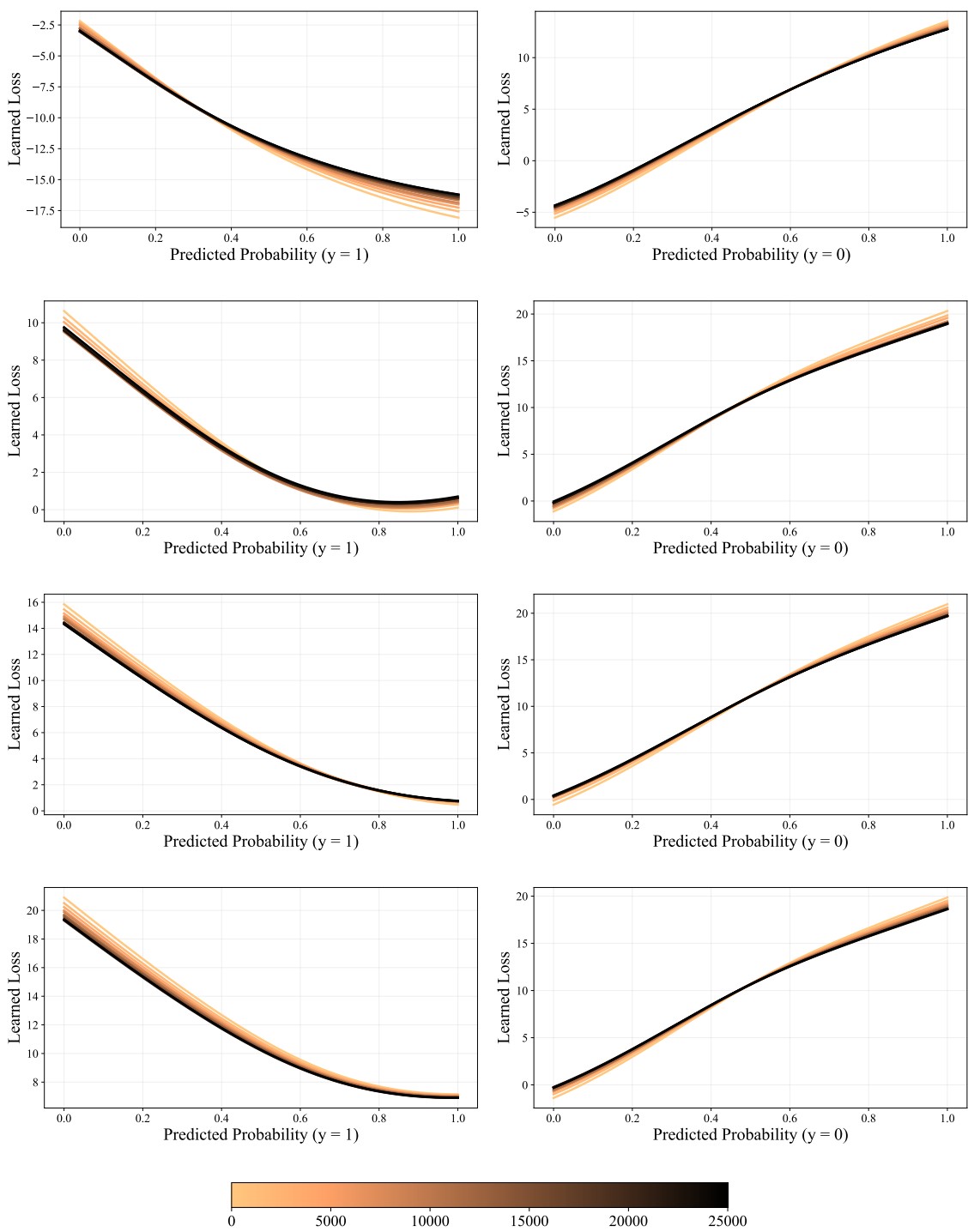

Figure 13: Loss functions generated by AdaLFL on the MNIST dataset, where each row represents a loss function, and the color represents the current gradient step.

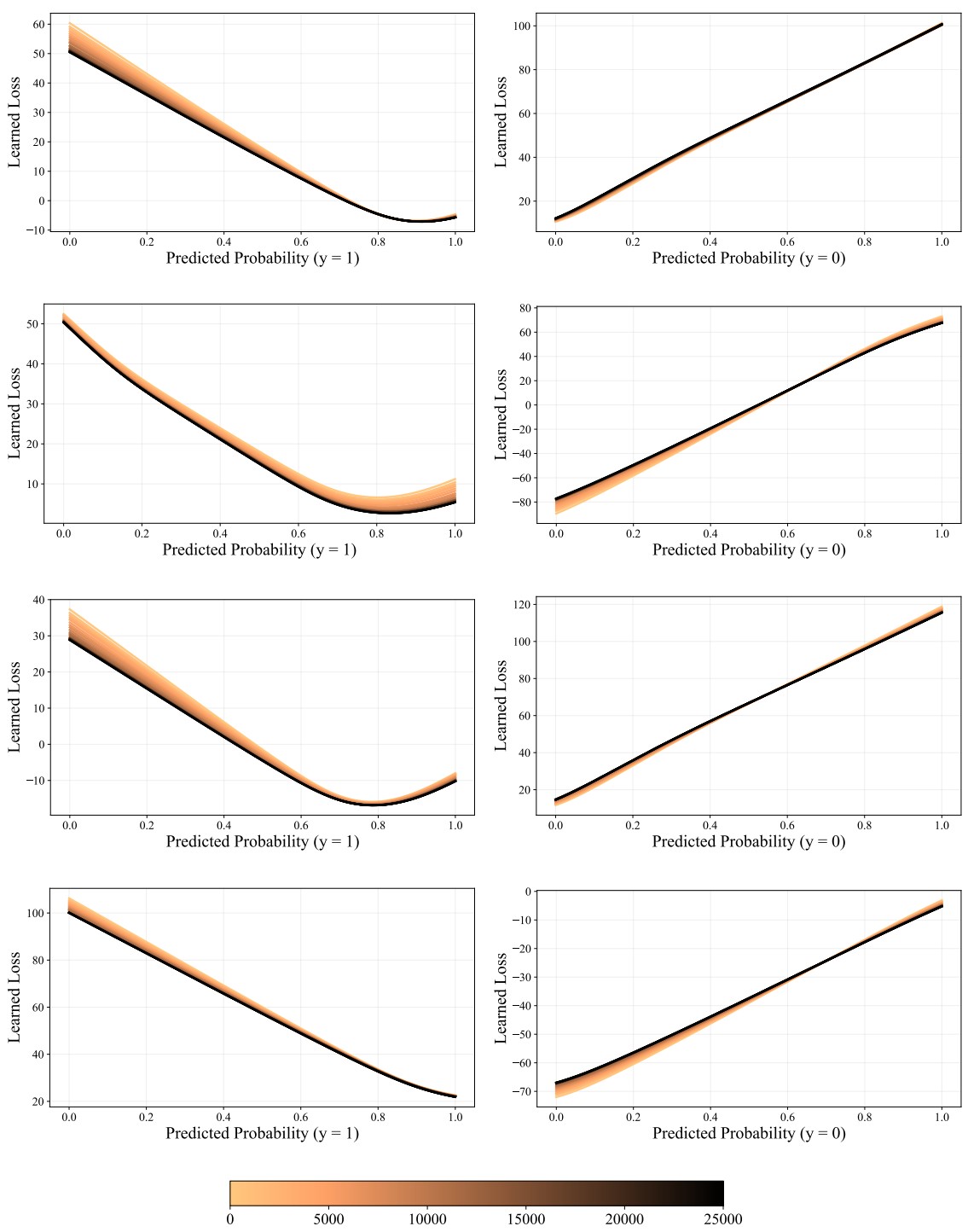

Figure 14: Loss functions generated by AdaLFL on the MNIST dataset, where each row represents a loss function, and the color represents the current gradient step.

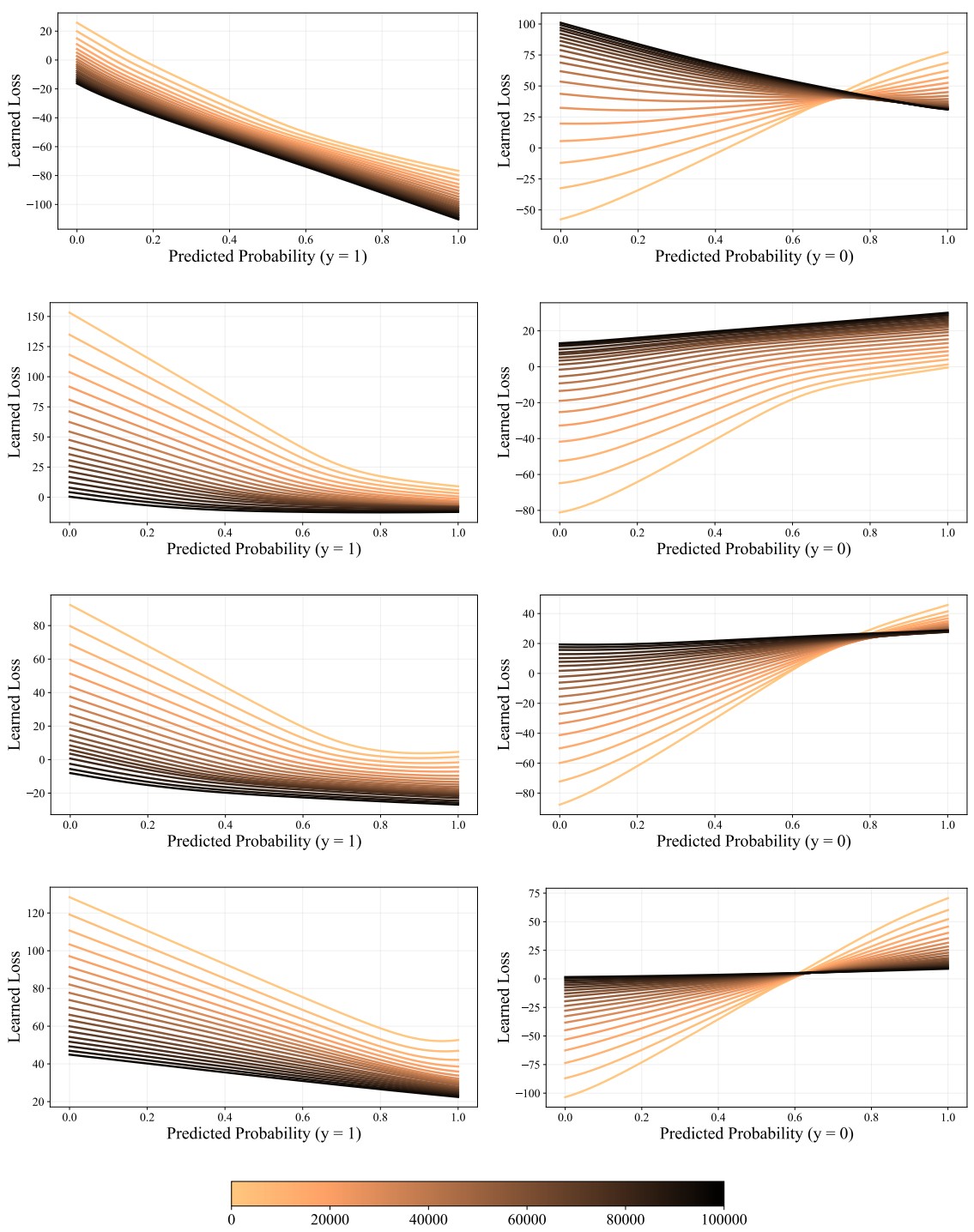

Figure 15: Loss functions generated by AdaLFL on the CIFAR-10 dataset, where each row represents a loss function, and the color represents the current gradient step.

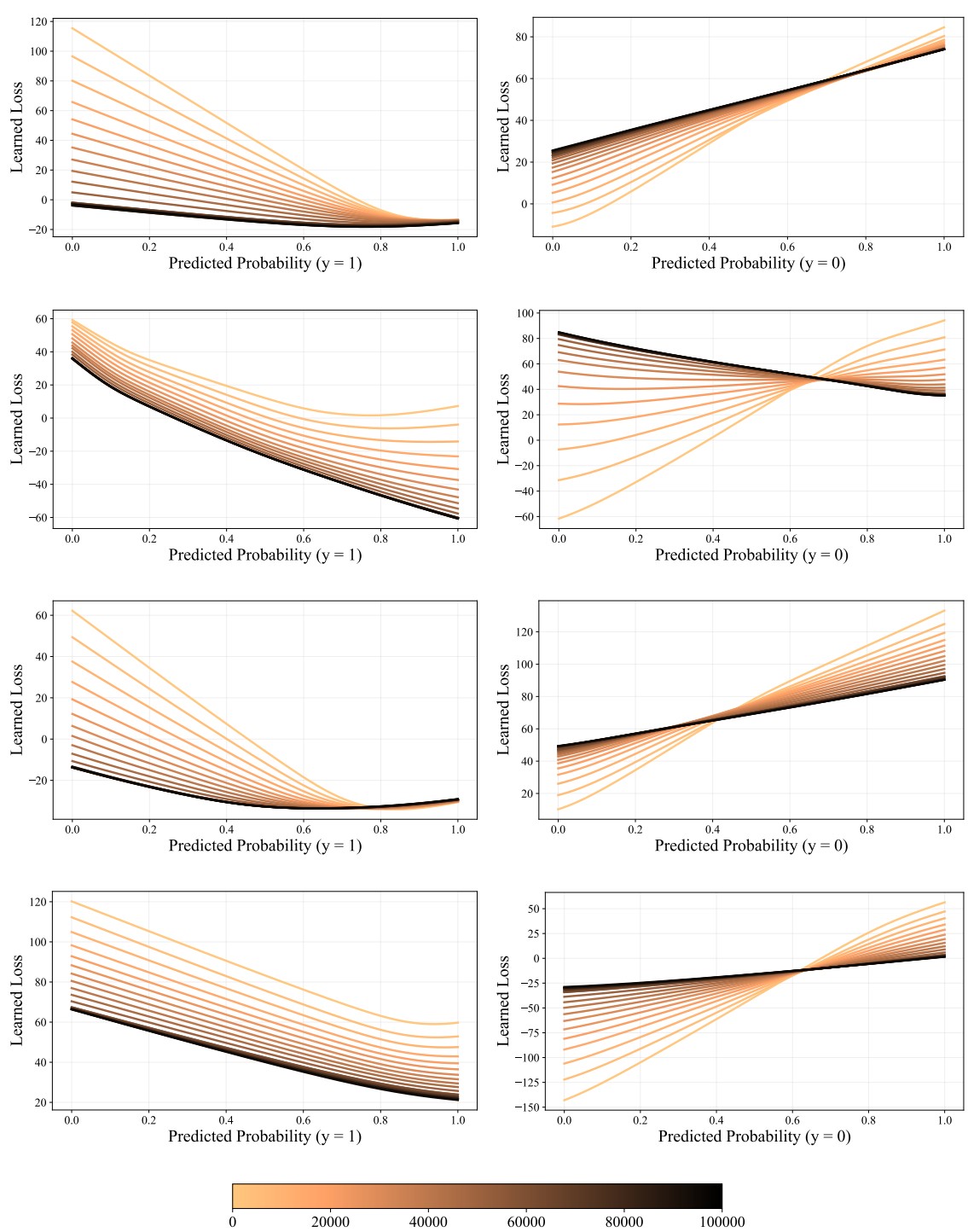

Figure 16: Loss functions generated by AdaLFL on the CIFAR-10 dataset, where each row represents a loss function, and the color represents the current gradient step.

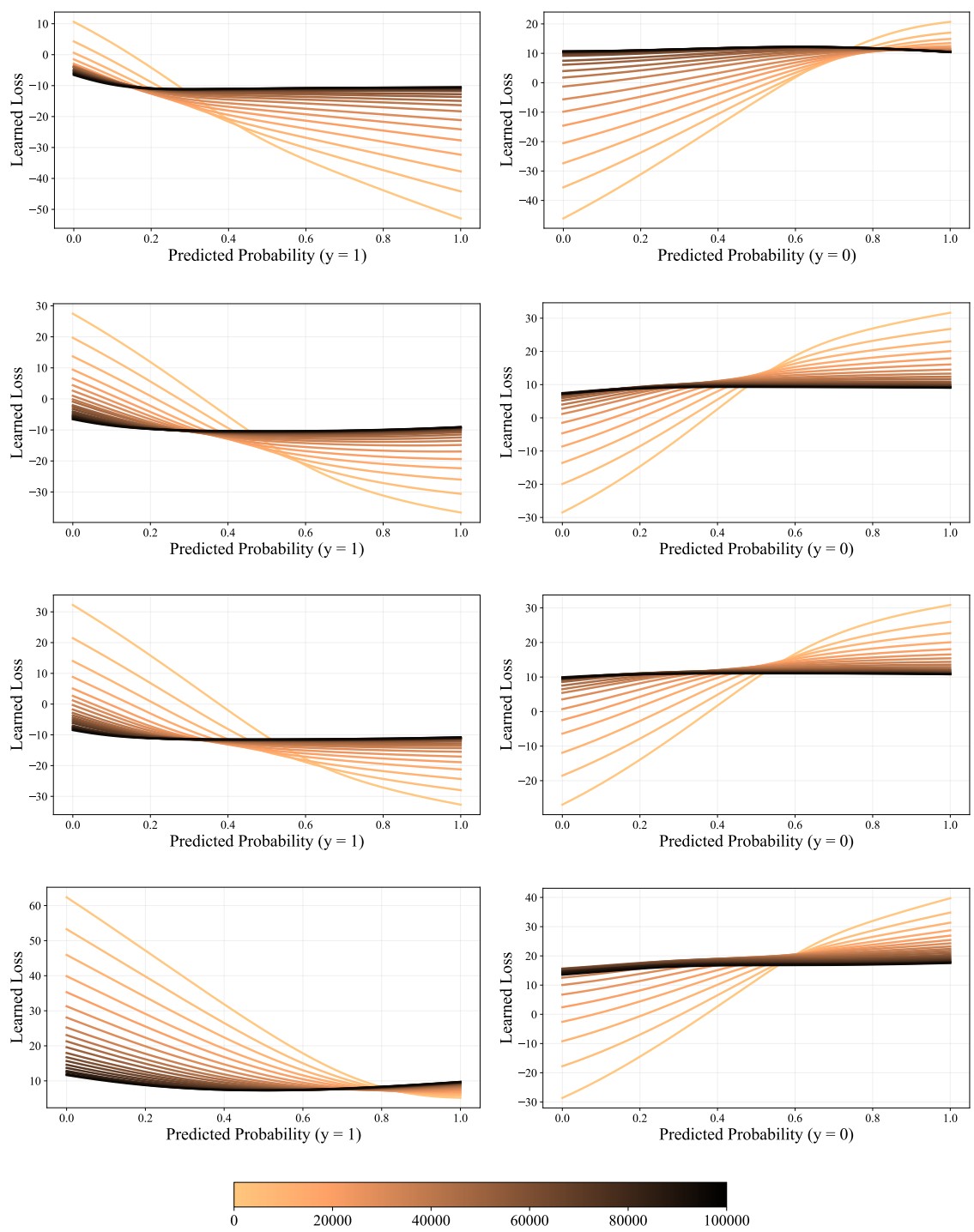

Figure 17: Loss functions generated by AdaLFL on the CIFAR-10 dataset, where each row represents a loss function, and the color represents the current gradient step.

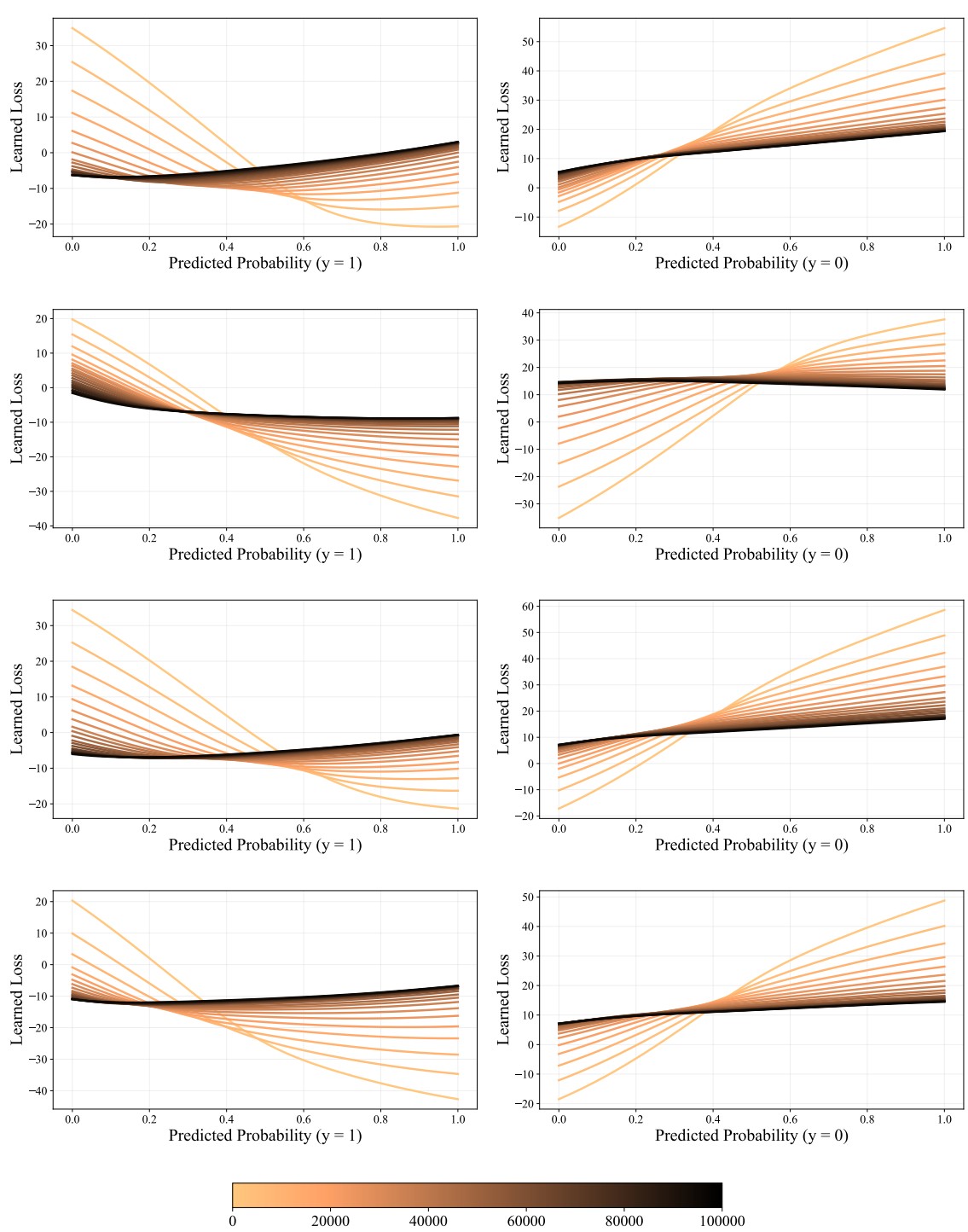

Figure 18: Loss functions generated by AdaLFL on the CIFAR-10 dataset, where each row represents a loss function, and the color represents the current gradient step.

