# OpenReview forum: "Meta-Learning Adaptive Loss Functions"
_TMLR — Accepted by TMLR_

### Review · Reviewer_4VXf · 2025-08-04

**Summary Of Contributions:**

This paper proposes a new meta-learning approach called Adaptive Loss Function Learning (AdaLFL), which automates the design of loss functions for machine learning models. Unlike previous "offline" methods that learn a static loss function before training the main model, which causes "short-horizon bias", AdaLFL learns and updates the loss function simultaneously and adaptively with the base model's parameters, adapting the shape and scale of the loss function throughout the training process. The learned loss function is represented as a small feed-forward neural network. The method consistently outperforms both traditional handcrafted loss function baselines and previous offline loss function learning techniques in experiments on a breadth of tabular regression and image classification tasks. It shows convergence over fewer training steps and better final inference performance. The paper addresses a flaw in previous neural network-based loss function designs that caused learned functions to become too flat and proposes a new smooth leaky ReLU activation function to prevent this. The adaptive nature of AdaLFL allows the learned loss function's scale to change over time, effectively tuning the learning rate schedule. The learned loss function can flatten or become parabolic at the end of training, which acts as a form of early stopping. In classification tasks, the loss function penalizes overconfident predictions, a behavior that resembles and dynamically adjusts an adaptive form of label smoothing.

**Audience:**

Yes

**Audience Explanation:**

The findings of this paper would be of significant interest to a broad segment of the TMLR audience, particularly those involved in meta-learning, hyperparameter optimization, and the fundamental theory of deep learning. This paper would interest an even wider audience if the methods were demonstrated on more modern architectures. For example, BERT-Small  $^1$ has fewer parameters than WRN-28-10, and SmolLM2-135M $^2$ also has an accessible size.

1. Turc, I., Chang, M. W., Lee, K., & Toutanova, K. (2019). Well-read students learn better: On the importance of pre-training compact models. arXiv preprint arXiv:1908.08962.

2. Allal, L. B., Lozhkov, A., Bakouch, E., Blázquez, G. M., ... & Wolf, T. (2025). SmolLM2: When Smol Goes Big--Data-Centric Training of a Small Language Model. arXiv preprint arXiv:2502.02737.

**Broader Impact Concerns:**

I have no broader impact concerns regarding this work. The paper focuses on a fundamental and general-purpose machine learning technique. The research advances the science of training models and does not present any obvious ethical issues that would necessitate a Broader Impact Statement beyond what is standard for foundational AI research.

**Claims And Evidence:**

Yes

**Claims Explanation:**

The paper presents experimental evaluation comparing AdaLFL against baselines and a leading offline loss function learning method (ML3). The experiments are conducted on a diverse range of tasks and models, including regression and classification datasets and various neural network architectures. The key claims summarized above are supported with quantitative results and qualitative discussion and analysis. The evidence presented is convincing and the narrative is clear overall. The authors have included ablation studies and comparisons to alternative methods (like meta-learning a learning rate) to strengthen their arguments, although as specified below, some of these should be moved from appendices to the main paper.

**Requested Changes:**

The following requested changes are roughly in order of importance.

Figure 3 is rather misleading, as the computational overhead of the offline initialization optimization and online bilevel optimization is not represented. Please at least factor in the steps for AdaLFL initialization, and even better use "FLOPs consumed" for the horizontal axes. The paper only discusses efficiency in the appendix, but this topic is important to discuss in the main paper so please move it to Section 5. Further, Figure 3 would be better be shown on a completely held-out set (disjunct from the newly created "validation" set), rather than the training data for each dataset. The subplots would be easier to parse with logarithmic vertical axes, as well as lowering the upper $y$-limit of most plots: cropping away some of the early high-error steps is acceptable in order to view the end of the curve more clearly. Finally, please include standard deviation as shaded bands in these plots, as Table 1 suggests that the spread of the results are rather overlapping.

I believe you use the newly created "validation" set for the outer optimization step. Since the "baseline" algorithm presumably does not have such a step, you should not hold-out this set from the training loop of baseline experiments. This is not specified in the paper, so I suggest to rerun the baseline experiments if necessary and explicitly state this dataset distinction in the text once it's added. Further, please explicitly describe the baseline algorithm(s) in Section 4, as I haven't found where it is specified if it is.

The meta-learned learning rate experiments are currently only discussed in the appendix, but should be moved to the main paper by merging Tables 1 and 3.

Adding experiments on transformer architectures would strengthen this work, but are not necessary for my recommendation for acceptance.

Please use textual citations (`\citet{}`) whenever the citation is a subject or object of the sentence. For example, "In (Bechtle et al., 2021), ReLU activations are..." should rather be "In Bechtle et al. (2021), ReLU activations are...".

---

> ### Author Response · Authors · 2025-08-26
>
> Thank you very much for taking the time out of your busy schedule to review our paper. We greatly appreciate your thoughtful comments on the significance of the work, as well as your recognition of the clear narrative and strong supporting evidence. We also value the detailed feedback you provided. We have highlighted our changes to the manuscript in red, and below, we address your comments point by point.
>
> # Responses:
>
> **1. RE: Learning Curves (Figure 3)**
>
> Thank you for your valuable suggestions for improving the presentation of Figure 3. We have updated Figure 3 to use a logarithmic y-axis and have included confidence intervals. Regarding the inclusion of offline meta-gradient steps, we believe this may introduce confusion, as these updates occur at the meta-level rather than the base-level. Furthermore, it is important to emphasize that the base model parameters are being reset at each iteration/gradient step in the initialization stage (see Algorithm 1, line 3). As for generating Figure 3 with a new held-out set, this would require re-running all experiments in the paper. Unfortunately, due to the limited response period, this is not feasible.
>
> **2. RE: Baseline Algorithm Description**
>
> We have added a new subsection, Section 4.1, as well as further expanding on the experimental setup in Appendix C to further discuss the methods used in our experiments. Regarding the use of the validation set, we have followed best practises of using an unseen held-out validation set for hyper-parameter tuning and model selection for the baseline and our proposed algorithm AdaLFL. In the case of the baseline algorithm this was used for manual tuning of the models’ hyperparameters, while for AdaLFL this was used for automatic tuning of the models hyperparameters (specifically the loss function). Therefore, we don’t consider the validation set used to be a new distinct set of examples, rather that we have just made use of an existing validation set during training to adapt the loss function, while the baseline uses it prior to training in a manual offline manner.
>
> **3. RE: Experiments on Transformers**
>
> Our experimental setup (i.e., models and benchmark datasets), were chosen following the established literature into meta-learning loss functions [1–4]. At present, our implementation relies on the Higher library for meta-optimization, which does not support distributed training. This limits us from scaling to larger models such as transformers. We fully agree, that extending our approach to additional model architectures (transformers, diffusion models, etc.) would be a valuable direction for future work.
>
> # Formatting and Minor Corrections:
>
> - **RE: Move Computational and Memory Complexity Section**: Following your suggestion we have moved this section from the appendix to section 5.2 and added further discussions about the computational and memory complexity.
> - **RE: Move Meta-Learning Learning Rates Section**: Thank you for your suggestion, we have moved this content into the main manuscript and merged Tables 1 and 3 and updated the analysis accordingly.
> - **Bechtle et al. Reference Formatting**: This has been updated in the new version of the manuscript, thanks!
>
> # References:
>
> - [1] Gonzalez, S., & Miikkulainen, R. Improved Training Speed, Accuracy, and Data Utilization through Loss Function Optimization.
> - [2] Gonzalez, S., & Miikkulainen, R. Optimizing Loss Functions through Multi-Variate Taylor Polynomial Parameterization.
> - [3] Bechtle, S., et al. Meta-Learning via Learned Loss.
> - [4] Raymond, C., et al. Learning Symbolic Model-Agnostic Loss Functions via Meta-Learning.

---

### Review · Reviewer_E7po · 2025-08-04

**Summary Of Contributions:**

This work addresses the challenge of handcrafting loss functions or learning them in an offline manner, both of which tend to perform well in early stages of training but not in later stages. Instead, the authors lift the assumption of a fixed loss function throughout training and propose online loss function learning through AdaLFL (Adaptive Loss Function Learning), a bilevel optimization algorithm that jointly optimizes task parameters and a loss network. The loss network is updated for each training iteration of the base task network, in addition to some "warm-start-type" initialization of the loss network. The paper empirically shows that models trained with AdaLFL exhibit faster convergence and improved inference performance compared to those trained with handcrafted or offline-learned loss functions, for several choices of datasets and model architectures. It also attempts to analyze the learned loss functions to interpret why they perform well. The overall ideas of the paper are clearly presented, although with a few typos and some ambiguous figure captions (listed in the Requested Changes).

**Additional Comments:**

Overall, I find the online loss learning paradigm a useful one and the authors' proposed algorithm interesting. However, the clarity of writing and soundness of its evaluation could be improved to help communicate its benefits.

**Audience:**

Yes

**Audience Explanation:**

Handcrafting loss functions manually is a time-consuming and challenging dark art that a lot of researchers have to deal with currently. I believe TMLR's audience, especially ML practitioners, would be interested in alternatives that perform well and do not have prohibitive computational overheads. A method like the online loss learning paradigm proposed by this work has the potential to improve researchers' efficiency as well their task performance.

**Broader Impact Concerns:**

The paper does not contain a Broader Impacts Statement. Authors should consider discussing the computational overhead that their method has.

**Claims And Evidence:**

No

**Claims Explanation:**

While the paper offers an interesting and intuitive idea, i.e. of online loss function learning, the experimental evidence lacks soundness in a few ways:
- It is unclear whether the better training curves of AdaLFL shown in Figure 3 are due to improved learning or more overfitting. Showing validation curves instead would improve this piece of evidence.
- The test performance gains are small in several cases, relative to the variance of the results. Hence, strong improvement due to AdaLFL is difficult to claim.
- The qualitative interpretation of the learned loss curves is definitely interesting, but there isn't much formalization or quantitative evidence to support claims about implicit early stopping or implicit label smoothing. That said, the implicit learning rate scheduling claim is well-supported by the change in the scale of the learned loss.

**Requested Changes:**

- The takeaway of Figure 1 is a bit unclear at the point in the paper at which it appears. In general, authors might consider stating each figure's takeaway in its caption. Is there any difference between Figure 1 and Figure 4, except that they show different examples?
- Please clarify whether the "gradient steps" referred to in the caption of Figure 1 refer to training iterations. What does the "predicted probability" refer to? Is it of the true class? The (y=0) and (y=1) labels are confusing since CIFAR is not a binary classification task. Also, does every "row" in the caption of Figure 1 refer to each curve?
- The authors state that the "principal goal of AdaLFL is to replace this conventional handcrafted loss function with a meta-learned adaptive loss function". However, given that AdaLFL depends on the task loss function to train the learned loss network, it doesn't quite replace it, but perhaps learns from it. I would recommend improving the wording sos as to avoid overstating claims.
- Does "Appendix 2.1" in section 2.2 refer to B.1? Authors might consider using `cref` to refer to different section of the paper
- The citation for Bechtle et al., 2021 after equation 3 seems to use `citep` but should probably use `citet`.
- Do authors mean "If $S_{inner} > 1$" towards the end of section 2.3?
- Can authors show validation curves instead of, or in addition to, the training curves in Figure 3, so that learning can be disambiguated from overfitting during training?
- Since there is space in the paper, the figures that are supposed to provide evidence for the implicit early stopping regularization and implicit label smoothing claims should appear in the main paper, as opposed to the appendix. Is it possible to quantify this effect and investigate whether it appears systematically across settings?
- What is the additional computational cost of using AdaLFL during training, including the "initialization" training it requires?

All the above requested changes would be critical to making the paper well-written and supporting its claims.

---

> ### Author Response · Authors · 2025-08-26
>
> Thank you for reviewing our paper, and for the detailed comments and thoughtful observations. We also appreciate your comments about the relevance of our research direction, and how it can potentially be used to improve researchers' efficiency in designing loss functions. We have highlighted our changes to the manuscript in red and below, we respond to your comments in detail.
>
> # Responses:
>
> **1. RE: Implicit Early Stopping Regularization and Label Smoothing**
>
> Following your suggestion, we have moved the figures illustrating implicit early stopping regularization and label smoothing from the appendix into the main manuscript (now in Figure 4). With respect to quantifying these effects, this remains challenging due to the black-box nature of neural networks. Prior work [1, 2] has analysed these behaviours theoretically, but only in the context of more interpretable, albeit less expressive, parameterizations such as cubic Taylor polynomials and genetic programming expression trees. While a formal treatment was not feasible in our setting, we believe that our qualitative results provide clear evidence that implicit early stopping regularization and label smoothing are common emergent behaviours agnostic of the loss function parameterization used.
>
> **2. RE: Training vs Validation Curves**
>
> Thank you for your suggestion. We agree that reporting the validation curves would typically be helpful for better disambiguating between learning and overfitting. However, in our special case, since the meta-learned loss function is optimized using the validation set, reporting validation curves could give a misleading and false impression by making our proposed method appear substantially better than the baseline (i.e., our method is actively optimizing the validation set in the outer optimization while the baseline and ML3 are not). Therefore, to not mislead readers we have decided to report the training learning curves instead. Importantly, the quantitative results in Table 1 clearly demonstrate that the improved training performance of our method generalizes to new unseen testing data. Therefore, it can be concluded that the results in Figure 3, is due to improved learning and training dynamics, and not just overfitting.
>
> # Clarifications:
>
> **1. RE: Figure 1 vs Figure 4**
>
> Both these figures are visualization of our adaptive loss functions. The intention of including Figure 1 at the start of the manuscript was to help aid readers intuitions early on by giving a concrete example of some adaptive loss functions so that they could get a better idea of what our paper aims to achieve prior to going into the technical content in Section 2, which may otherwise feel overly abstract. The loss functions in Figure 4 are also specifically chosen to highlight the implicit meta-learning behaviours of our proposed method, with corresponding analysis in Sections 5.4-5.7
>
> **2. RE: Gradient Steps and Predicted Probability (Figure 1)**
>
> Regarding the visualizations of the loss functions (Figures 1, 4, 5, 8–17), the term gradient steps refers to training iterations, i.e., each update to the base model parameters $\theta$. The predicted probability corresponds to the model’s output probabilities, $f_{\theta}(x) \in [0,1]^{\mathcal{C}}$. In the figure, the expressions in parentheses indicate whether the plot shows the target loss (left, which penalizes deviation from 1) or the non-target loss (right, which penalizes deviation from 0). Since the loss is applied output-wise (Section 2.2, Equation 3), it can be interpreted as a binary classification task applied independently to each of the $\mathcal{C}$ outputs. To reduce confusion, we have updated the notation from $y=1$ to $y_i=1$ to explicitly communicate that the labels correspond to individual output indices rather than a global binary label.
>
> **3. RE: Computational Cost of AdaLFL**
>
> Information about the computational cost of the proposed method can be found in Section 5.2 of the manuscript. In this section we report the run-time of the baseline, MetaLR, ML3, and our method, AdaLFL. We also discuss three key reasons why the computational overhead is admissible. The extra computational overhead used by our proposed algorithm comes from (1) taking an additional forward propagation step on the validation set, and (2) backpropagating through the one-step trajectory. These operations are shown in the outer optimization (yellow) in Figure 2.

---

> ### Author Response · Authors · 2025-08-26
>
> # Formatting and Minor Corrections:
>
> - **Rephrasing Algorithm Claim**: We agree with your assessment and have reworded the claims made in that paragraph to more accurately reflect the proposed algorithm.
> - **Figure Caption Clarity**: Thank you for your suggestion, we have gone through and updated the figure captions to better highlight the takeaway the reader should have.
> - **Appendix Section Referencing**: Reference now points to the correct appendix section.
> - **Bechtle et al. Reference Formatting**: We have updated reference to use \citet{}.
> - **Section 2.3, $S_{inner} < 1$**: Thank you for spotting this, we have corrected it in the final manuscript.
>
> # References:
>
> - [1] Gonzalez, S., & Miikkulainen, R. Effective Regularization through Loss-Function Meta-Learning.
> - [2] Raymond, C., et al. Learning Symbolic Model-Agnostic Loss Functions via Meta-Learning.

---

> > ### Comment · Reviewer_E7po · 2025-09-02
> > **Response to Author Clarifications**
> >
> > Thank you to the authors for providing clarifications and updates to the paper!
> >
> > Most of my concerns are addressed, however I still believe that reporting loss curves on data other than training data would provide stronger evidence, although I understand the authors' concern around using validation data that has already been used to optimize AdaLFL. The solution to this would be to split the data into 3 sets: `train` for the inner optimization, `val` for the outer optimization of the meta-learning set-up, and `eval` for unseen data on which evaluation metrics can be tracked during training. This would be great to have to improve the paper. Reporting test performance doesn't quite make up for this since it doesn't tell us much about the change in metrics throughout training.

---

> > > ### Author Response · Authors · 2025-09-03
> > >
> > > Thank you again for taking the time to review our work, and thank you for noting that most of your concerns have been addressed!
> > >
> > > Following prior literature, we initially chose the current partitioning strategy (train/val/test) to ensure experimental comparability across papers. However, we fully agree that introducing an additional evaluation set would have helped to better understand and provide insight into generalization throughout the training process. Unfortunately, this new experimental design would require us to rerun all experiments, which is unfortunately not feasible due to time constraints. If you feel that a smaller, targeted set of experiments using this newly proposed partitioning would be sufficient to address your concern, we would be happy to conduct and include those results (i.e., rerun experiments on MNIST or one of the CIFAR-10 models using the new partition strategy).

---

> > > > ### Comment · Reviewer_E7po · 2025-09-03
> > > >
> > > > I understand re-running everything may not be feasible due to time constraints. Since this would be a nice-to-have, a reasonable middle-ground would be to run this on one of the datasets / settings of your choice (as authors mentioned) and confirm that the results are as expected.

---

> > > > > ### Author Response · Authors · 2025-09-15
> > > > >
> > > > > Apologies for the delay! We have added additional experiments to Appendix D.5, where we visualize the learning curves throughout the training process on MNIST + LeNet-5. The performance is evaluated on an out-of-sample evaluation set distinct from the training set used to update the model parameters $\theta$, and from the validation set used to update the meta-parameters $\phi$. The results show that the proposed method improves both the convergence and generalization of the model compared to the baseline cross-entropy loss, as well as ML3 and MetaLR, as measured by an out-of-sample set of examples.
> > > > >
> > > > > We hope this new section has provided further empirical evidence of the enhanced generalization throughout training of our proposed method. Please let us know if you would like us to perform any further experiments or analysis.

---

> > > > > > ### Comment · Reviewer_E7po · 2025-09-15
> > > > > >
> > > > > > Thank you for the update!

---

### Review · Reviewer_NvPj · 2025-08-13

**Summary Of Contributions:**

The paper proposes Adaptive Loss Function Learning (AdaLFL), an online meta-learning method that updates a learned loss function after each base model update to address the short-horizon bias in prior offline approaches (e.g., ML3). A small feed-forward network parameterizes the loss, and smooth leaky ReLU activations are used to avoid flatness. Experiments on several datasets show improved convergence and final performance over cross-entropy and ML3.

Strengths:

1. Addresses a well-motivated limitation of offline loss function learning.
2. Experimental evaluation covers multiple datasets and architectures.
3. Some qualitative analysis of learned loss functions.

Weaknesses:

1. Method novelty is incremental; largely an adaptation of existing offline approaches to an online setting.
2. Computational and memory cost of unrolled differentiation not addressed; scalability to large models unclear.
3. Initialization stability issues (e.g., CIFAR-100) remain unresolved.
4. Experiments are on relatively small-scale benchmarks; unclear real-world applicability.

**Audience:**

Yes

**Audience Explanation:**

Loss function learning is a niche area, and the presented method is an incremental variation without compelling theoretical insight or large-scale validation.

**Broader Impact Concerns:**

No major ethical concerns, but the lack of scalability and limited evaluation on larger datasets.

**Claims And Evidence:**

No

**Claims Explanation:**

1. While the reported numbers show some improvement, the gains are often small, inconsistent across datasets, and not convincingly tied to the proposed online adaptation mechanism.

2. The paper does not sufficiently analyze why AdaLFL outperforms baselines in certain settings but fails to offer stable benefits in others.

3. What's more, the introduction of the hyperparameter of Eq 4 make the algorithm more sensitive and I doubt the generation of the algorithm.

**Requested Changes:**

1. Eq 4 list the formulation of loss function, I doubt the sensitivity of the hyperparapmeter. It is already a meta learning algorithm to learn hyper parameter, you add second order hyper parameters, which make the algorithm hard to tune. Is it correct? I believe a fix form of \phi can be better.

2. Provide a thorough computational/memory cost analysis for large models and larger datasets.

3. Analyze failure cases and stability issues in more depth.

---

> ### Author Response · Authors · 2025-08-26
>
> Thank you for taking the time out of your busy schedule to review our paper. We greatly appreciate your feedback. Please let us know if you have any more questions and we will do our best to get back to you promptly. We have highlighted our changes to the manuscript in red, and below we have responded to your comments in detail.
>
> # Responses:
>
> **1. RE: Second-Order Hyperparameter Sensitivity**
>
> In our experiments, the only hyper-parameter that showed notable sensitivity was the meta level learning rate $\eta$. We briefly discussed this in Appendix C. Specifically, in the online adaptation setting, if $\eta$ is set too high, the learned loss function can change too abruptly after each update, leading to oscillations in training dynamics. However, as long as $\eta$ is kept below this threshold, performance remains stable and consistent. To further illustrate this, we ran additional experiments on MNIST with LeNet-5 using $\eta \in \\{ 10^{−1}, \dots, 10^{-6} \\}$. These results have been included in Appendix D.3, Table 6, with supporting discussion on second-order hyperparameter sensitivity. The results show that AdaLFL consistently outperforms the baseline across a wide range of $\eta$ value, with degradation in performance only beginning when $\eta$ becomes excessively large.
>
> With regard to the reviewer’s concern about second-order hyperparameters, we would like to clarify that AdaLFL meta-learns the loss function parameters $\phi$, but this does not introduce an additional layer of manual hyperparameter tuning beyond what is already common and standard in meta-learning methods [1, 2, 3]. While one could adopt a fixed form $\phi$, this would greatly reduce adaptability. Our empirical results in Table 1 confirm this, as it shown that our adaptive loss function learning algorithm has consistent performance gains on the testing set over both the baseline and ML3, where the loss function is held static during training.
>
> **2. RE: Failure Modes**
>
> We believe we have sufficiently addressed key failure modes present in our algorithm. Specifically, in Section 2.2, we discuss a previously unknown (and very consequential) phenomenon where the loss function would unintentionally begin to flatten causing learning to prematurely converge. We identified that this issue was caused by using a bounded output activation function and subsequently resolved the issues by proposing a new output activation function, the smooth leaky ReLU. Subsequently a qualitative analysis of the problem was performed in Appendix B.1 and B.2, and a quantitative analysis in Appendix D.2.
>
> Regarding initialization issues on CIFAR-100, this is due to the first step meta-gradient obtained by ML3 not providing sufficient information to shape the initial state of the learned loss function. Despite this issue with ML3, proposed by [7], our method was able to adapt throughout the learning process and decreased the error rate compared to the baseline cross-entropy loss by 7.58%, which is a significant improvement in performance.
>
> **3. RE: Extensions to Larger Datasets**
>
> Our benchmark datasets were chosen following the established literature into meta-learning loss functions [5–8]. At present, our implementation relies on the Higher library for meta-optimization, which does not support distributed training. This limits us from scaling to larger models; however, we fully agree, that extending our approach to larger would be a valuable direction for future work.
>
> **4. RE: Proposed Method Performance**
>
> Regarding the performance of our proposed method, the results shown in Table 1, demonstrate that our algorithm can make consistent gains over the baseline (using handcrafted loss functions) and offline loss function learning method ML3, across all 7 datasets and 8 model architectures. The gains may appear to be relatively small; however, this is due to the use of heavily tuned baselines, which we spent considerable time tuning (we report significantly better results for ML3 then the authors did in their original paper [4]).
>
> # References:
> - [1] Finn, C., et al. Model-Agnostic Meta-Learning for Fast Adaptation of Deep Networks.
> - [2] Baydin, A. G., et al. Online Learning Rate Adaptation with Hypergradient Descent.
> - [3] Hospedales, T., et al. Meta-Learning in Neural Networks: A Survey.
> - [4] Bechtle, S., et al. Meta-Learning via Learned Loss.
> - [5] Gonzalez, S., & Miikkulainen, R. Improved Training Speed, Accuracy, and Data Utilization through Loss Function Optimization.
> - [6] Gonzalez, S., & Miikkulainen, R. Optimizing Loss Functions through Multi-Variate Taylor Polynomial Parameterization.
> - [7] Bechtle, S., et al. Meta-Learning via Learned Loss.
> - [8] Raymond, C., et al. Learning Symbolic Model-Agnostic Loss Functions via Meta-Learning.

---

### Decision · Action_Editor_SFjF · 2025-09-25

**Recommendation:** Accept with minor revision

**Additional Comments:**

I request that the authors prepare the final version based on the latest revision.
Please double-check the changes the reviewers requested.

I encourage the authors to follow [the Math section of this guide](https://www.jmlr.org/format/formatting-errors.html).

On page 3: there is a typo: $\ell \phi$. $\mathcal{C}$ might be undefined in Eq. (3).

In Algorithm 1, the loop variable might be $t \in \{0, \dots, S_{\text{inner}} - 1 \}$ if the authors wanted $S_{\text{inner}} = 1$ to mean one iteration. The same goes for other loops.

**Audience:**

Yes

**Audience Explanation:**

The reviewers agree that at least some people would be interested in the findings, although one of them mentioned that the audience might be limited.

**Claims And Evidence:**

Yes

**Claims Explanation:**

The main claims of the paper are as follows.
- Existing offline loss function learning methods were only able to take early-stage training into account.
- The proposed method addresses this weakness by adaptively training the loss function model together with the predictive model until convergence, which is expected to improve the quality of the learned loss function even in later stages of training.
- Experiments with several datasets and model architectures show that the proposed method often reaches convergence in fewer training steps (Figure 3) and improves prediction performance compared with handcrafted or loss functions learned offline.
- The method also adjusts the scale of the loss function during the training, which serves as adaptive learning rate scheduling. The change of the scale of the learned loss was indeed observed in the experiments.
- The plots of Figure 4 and Appendix D.6 show that the adaptive loss functions penalize overconfident predictions in later stages, which acts as early stopping.
- The paper also proposes a new smooth leaky ReLU activation function to address flat loss functions observed with a conventional activation (Figure 5).
- Additional experiments such as hyperparameter sensitivity, computation costs, evaluation with held-out data were provided to address the comments raised during the discussion period.

Overall, the claims are fairly well-supported by empirical results. The reviewers also mention that the paper presents the ideas in a clear manner.